# Optogenetic activation of visual thalamus generates artificial visual percepts

Jing Wang[1,2], Hamid Azimi[1], Yilei Zhao[1], Melanie Kaeser[1], Pilar Vaca Sánchez[1], Abraham Vazquez-Guardado[3], John A Rogers[4], Michael Harvey[1], Gregor Rainer[1]*

[1]Department of Medicine, University of Fribourg, Fribourg, Switzerland; [2]Department of Neurobiology, School of Basic Medical Sciences, Nanjing Medical University, Nanjing, China; [3]Department of Material Science and Engineering, Northwestern University, Chicago, United States; [4]Querrey Simpson Institute for Bioelectronics, Northwestern University, Evanston, United States

**Abstract** The lateral geniculate nucleus (LGN), a retinotopic relay center where visual inputs from the retina are processed and relayed to the visual cortex, has been proposed as a potential target for artificial vision. At present, it is unknown whether optogenetic LGN stimulation is sufficient to elicit behaviorally relevant percepts, and the properties of LGN neural responses relevant for artificial vision have not been thoroughly characterized. Here, we demonstrate that tree shrews pretrained on a visual detection task can detect optogenetic LGN activation using an AAV2-CamKIIα-ChR2 construct and readily generalize from visual to optogenetic detection. Simultaneous recordings of LGN spiking activity and primary visual cortex (V1) local field potentials (LFPs) during optogenetic LGN stimulation show that LGN neurons reliably follow optogenetic stimulation at frequencies up to 60 Hz and uncovered a striking phase locking between the V1 LFP and the evoked spiking activity in LGN. These phase relationships were maintained over a broad range of LGN stimulation frequencies, up to 80 Hz, with spike field coherence values favoring higher frequencies, indicating the ability to relay temporally precise information to V1 using light activation of the LGN. Finally, V1 LFP responses showed sensitivity values to LGN optogenetic activation that were similar to the animal's behavioral performance. Taken together, our findings confirm the LGN as a potential target for visual prosthetics in a highly visual mammal closely related to primates.

*For correspondence:
gregor.rainer@unifr.ch

**Competing interest:** The authors declare that no competing interests exist.

## Editor's evaluation

This important study shows that tree shrews can detect optogenetic stimulation of the lateral geniculate nucleus (LGN) after training detection of visual stimuli. The solid evidence links optogenetic stimulation of the LGN to behavioral detection and neurophysiological responses. This article is potentially of interest to neuroscientists and clinicians working on the visual system and visual prostheses.

## Introduction

Vision loss is widespread and rapidly progressing worldwide due to an aging population, with over 45 million people suffering from blindness in 2020 and over 55 million cases forecast for 2050 (**Bourne et al., 2021**). For many eye diseases, including glaucoma, macular degeneration, and retinitis pigmentosa, there are no available treatments, and they progressively lead to complete loss of vision (**Roska and Sahel, 2018**). It has been shown that prosthetic electrical stimulation in visual structures can produce visual percepts, in the form of phosphenes, and at least partially restore visual function (**Mills et al., 2017**). To date, the most advanced of these approaches are retinal implants, with sub- and

epiretinal approaches that have both been extensively tested in human patients. Despite considerable progress, the percepts generated by these prostheses remain far from natural vision, and many patients stop using their implants and often report limited improvements in quality of life (*Ho et al., 2015*). It is interesting to note that phosphenes tend to vary substantially in terms of their appearance between patients, but they do not change much with time (*Erickson-Davis and Korzybska, 2021*). This suggests that differences in percept may result not from neural plasticity but rather from variations in device implantation such that slightly different parts of the retinal circuitry are activated by the implant in each individual. Indeed, the retina is a highly complex computational engine containing dozens of distinct, specialized cell types (*Masland, 2012*), and electrical stimulation is not cell-type specific and can trigger activation in any neurons within a defined volume around the electrode tip. Another pertinent limitation relates to the size of the retinal implants, which can cover only up to about 20° of visual angle such that patients must make head movements to repeatedly scan the environment during locomotion to detect obstacles, walls or doors. The interpretation of phosphenes generated by retinal prosthetic devices generally does not becomes effortless or intuitive, but continues to require considerable cognitive resources and effort. Nevertheless, technical advances under development (*Chenais et al., 2021*) may certainly enhance the usefulness of retinal implants in the future.

The primary visual cortex (V1) has also been investigated as a target for electrical visual prosthetics due to attractive features such as its large surface area and could enable highly precise representations of the visual environment using suitable stimulation arrays. Early work in this area has demonstrated proof of principle (*Brindley, 1982*; *Dobelle et al., 1976*) that electrical activation of V1 can elicit phosphene percepts, and there has been resurgent interest in cortical visual prostheses in recent years (*Lewis et al., 2015*; *Bosking et al., 2017*). For example, it was recently shown that monkeys could detect, and make saccades to, V1 stimulation at particular retinotopic sites, as well as interpret multielectrode stimulation in the context of a learned visuomotor association task (*Chen et al., 2020*). This work relied on the implantation of over 1000 penetrating electrodes in multiple arrays in visual cortex, highlighting the technical challenge of achieving broad coverage for optimizing visual prosthetic stimulation. In another recent study, blind human subjects readily identified sequential electrical activation on a rectangular grid of flexible electrodes spaced 1 cm apart that represented letter symbols, whereas simultaneous electrical activation was much harder to comprehend (*Beauchamp et al., 2020*). This latter finding suggests more generally that sequential patterned activation across multiple sites could be highly useful in producing more coherent percepts. Numerous issues remain however in relation to cortical electrical visual prostheses, including problematic surgical access to the large part of V1 that is buried in the calcarine sulcus and particularly for penetrating electrodes, potential damage caused by the electrodes during implantation, long-term stability of the electrode-neural interface, and risk of triggering seizures due to the highly recurrently connected nature of the cortex. An advantage of V1 prosthetics is that they are more general and can be applied also in eye disorders where the retinal ganglion cells are damaged, such as glaucoma. This same advantage would also apply to the lateral geniculate nucleus (LGN) of the thalamus, where the ganglion cell axons synapse onto the thalamocortical neurons whose axons deliver visual information to cortex (*Meikle and Wong, 2022*; *Pezaris and Eskandar, 2009*). Indeed, electrical activation of LGN has been shown to activate V1 neurons (*Panetsos et al., 2011*), and LGN activation can be detected by monkeys, who are able to saccade reliably to the corresponding visual field location (*Pezaris and Reid, 2007*). In terms of prosthetics, the LGN has another advantage in that it is a six layered retinotopically organized nucleus where information pathways are segregated and could therefore be specifically targeted (*Conway and Schiller, 1983*; *Sherman et al., 1975*). However, the LGN is a deep brain structure far from the cortical surface and has to date not been much studied in terms of prosthetic applications.

Gene therapy and optogenetics have also brought novel approaches for combating eye diseases. For example, the light-sensitive channel-rhodopsin protein was expressed in retinal ganglion cells of a patient suffering from photoreceptor loss due to retinitis pigmentosa, leading to recovery of some visual function using suitable illumination via goggles (*Sahel et al., 2021*). Optogenetics also opens another avenue for vision restoration, combining viral delivery to activate elements of the visual pathway with patterned laser or LED light stimulation to produce visual percepts (*van Wyk et al., 2015*). The cell-type specificity of optogenetics is likely to confer advantages compared to electrical stimulation as only cell types of interest can be activated. Both cortex and LGN, for example, contain numerous inhibitory interneurons, whose activation may hamper the efficacy of prosthetic stimulation.

In this study, we focus on the LGN as a potential target of optogenetic restoration of vision using optogenetic activation of thalamocortical projecting neurons. We employ ChR2 coupled to the CamKIIα promotor as it is known to induce activation as well as plasticity in excitatory neurons in the mouse (*Lee et al., 2009*; *Shibata et al., 2021*). We characterize opsin expression in LGN and V1 and assess the impact of opsin activation on both LGN circuit activity, and behavioral detection task performance linked to optogenetic activation. We use tree shrews as an animal model in this study, as they are a diurnal mammalian species closely related to primates that have multiple advantages for the translational study of visual prosthetics. For example, tree shrew V1 has a large surface area of 4000 mm² and exhibits orientation columns and shares many other aspects of functional organization with primates (*Fitzpatrick, 1996*; *Khani et al., 2018*; *Mohan et al., 2022*; *Veit et al., 2011*). Their retina is composed largely of cones (*Muller et al., 1989*), which is useful for prosthetic work as humans tend to illuminate their environments; they are a diurnal mammal that is highly reliant on vision (*Emmons, 2000*) and can readily be trained on cognitive visually based behavioral tasks (*Mustafar et al., 2018*; *Schumacher et al., 2022*).

## Results

### Anatomical and functional validation of viral transfection

We performed immunohistochemistry for CaMKIIα and show robust staining of cell bodies in all LGN layers (see *Figure 1A* for an example animal and *Figure 1—figure supplement 1* for additional animals), which indicates that this promotor could be a useful target for optogenetic activation of tree shrew LGN neurons. We complemented this with parvalbumin (PV) immunohistochemistry, which is useful for delineating LGN laminar structure (*Figure 1A*, center panel). We observed that PV expression varied somewhat across LGN layers, with particularly strong expression in layer 4 and weaker expression in layers 3 and 6. These observations are consistent with previous reports in the tree shrew documenting paler parvalbumin staining in layers 3 and 6 (*Diamond et al., 1993*; *Usrey et al., 1992*), and support the notion that laminar LGN structure is more readily discernible using PV rather than CamKIIα immunohistochemistry. We injected adeno-associated (AAV2) virus that contained a construct for the light-sensitive ion channel ChR2 and the enhanced red fluorescent protein mCherry under the control of the CamKIIα promotor into the tree shrew LGN of animals used for behavioral or electrophysiological studies. We found robust expression of CamKIIα-ChR2 within LGN layers in the vicinity of the injection site including on cell bodies, for example, targeting inner LGN layers 1, 2, and 3 for a medial LGN injection (*Figure 1B*, see *Figure 1—figure supplement 2* for additional animals). We also observed clear labeling of thalamocortical axons in layers 2/3 and 4 of V1 (see *Figure 1C* for an example animal and *Figure 1—figure supplement 3* for additional animals). In fact, both V1 cortical recipient layers contain axons expressing CamKIIα-ChR2, which highlights that neurons in both LGN layer groups 3/6 as well as 1/2/4/5 robustly express the light-sensitive ion channel as these project respectively to V1 layers 3 and 4 (*Van Hooser et al., 2013*). Note that in macaques, CamKIIα is expressed mostly in interlaminar LGN layers, which are composed of konio-type relay neurons (*Benson et al., 1991*; *Klein et al., 2016*), such that the CamKIIα-positive population activates a more restricted group of thalamocortical projections in macaques compared to tree shrews.

For functional validation, we performed terminal experiments under isoflurane anesthesia using optrodes that allowed simultaneous recording of neural activity and activation of ChR2 using an optic fiber within the LGN. Results for an example recording site are shown in *Figure 2A*, illustrating entrainment of LGN spiking activity to light transients at several frequencies of stimulation. Frequency analysis of the evoked spike trains reveals clear peaks at the stimulation frequency, as well as harmonics. Note that at the 80 Hz stimulation frequency, the neural response adapts rapidly, and the neuron tends to respond only to the onset of stimulation; a finding that is expected due to ChR2 channel dynamics, but indeed also resembles the activation profile to high-frequency visual flicker (*Veit et al., 2011*). The LGN can be activated in tonic or burst mode, which favors either stimulus fidelity or detectability respectively (*Adams et al., 2002*). Action potentials in both cases, however, convey stimulus-specific information to the cortex (*Ortuño et al., 2014*; *Reinagel et al., 1999*). We therefore examined to what degree the optogenetic activation triggered bursts in our recordings. We recorded from 120 single units in the LGN and defined a burst as two or more action potentials with inter-spike intervals of <4 ms, and preceded by at least 50 ms without spiking activity. For the example neuron (*Figure 2A*),

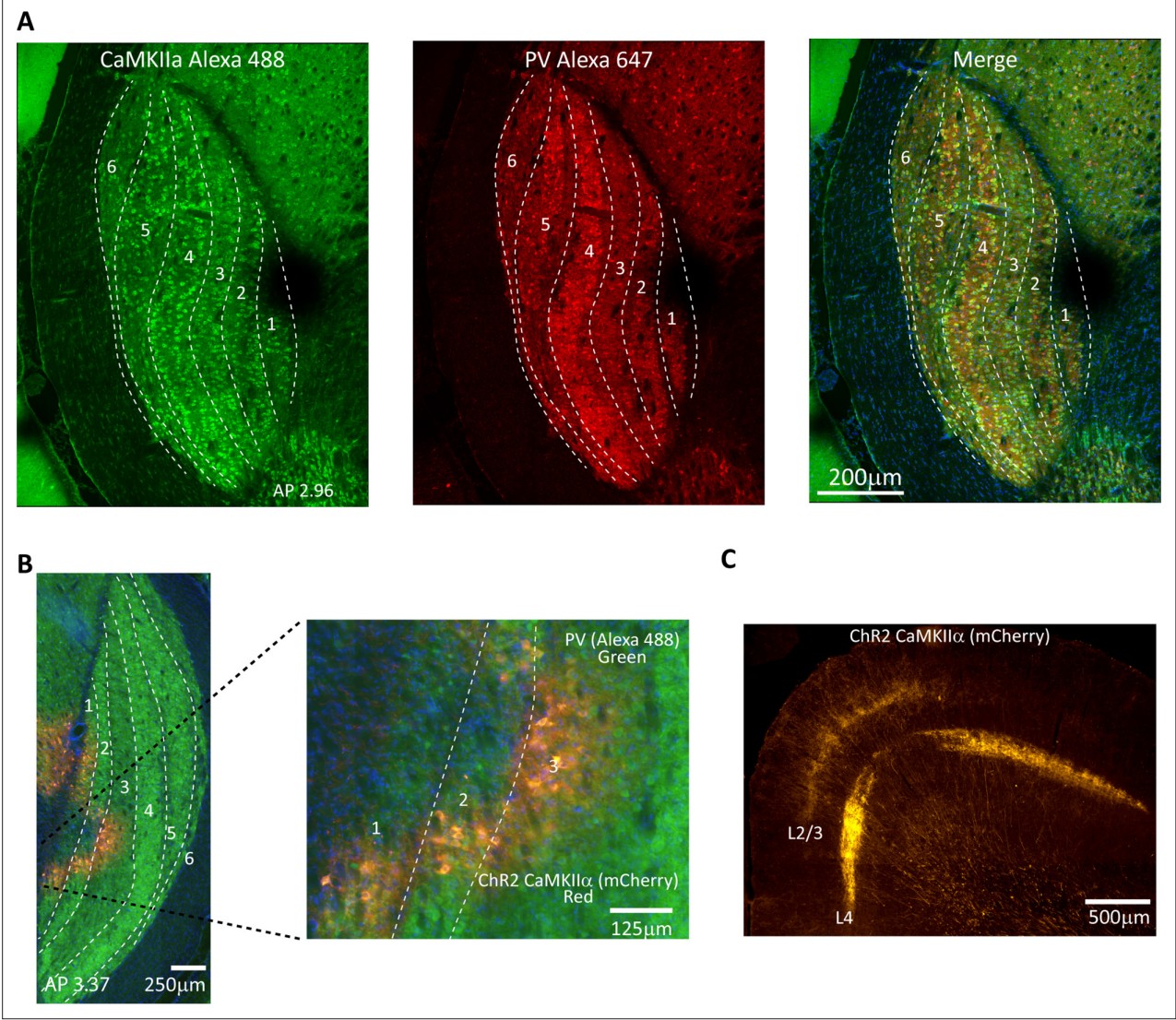

**Figure 1.** Validation of CaMKIIα and ChR2 expression in tree shrew lateral geniculate nucleus (LGN). (**A**) Confocal images of tree shrew LGN immunostained for both *CaMKIIα* (left) and parvalbumin (center), with the merge shown at right. Dashed lines indicate laminar boundaries. Note that CaMKIIα is found throughout LGN laminae as well as in the interlaminar zones. AP coordinates are from the interaural line. (**B**) Epifluorescent image immunostained for parvalbumin, green, revealing LGN layers, and showing viral expression (mCherry, red) in LGN layers 1–3. (**C**) Axonal projection patterns in V1 from viral transfected cells in the LGN. Note prominent projections to both granular and superficial layers.

The online version of this article includes the following figure supplement(s) for figure 1:

**Figure supplement 1.** Immunohistochemistry for CaMKIIa and parvalbumin (PV) in lateral geniculate nucleus (LGN).

**Figure supplement 2.** ChR2 expression in CaMKIIα neurons in tree shrew lateral geniculate nucleus (LGN).

**Figure supplement 3.** ChR2 expression in CaMKIIα thalamocortical axons in tree shrew V1.

bursts were rarely triggered and burst percentage of total spikes amounted to 0.7 ± 0.3%, with other recorded neurons also in this range (n = 7, range: 0.03–3.0%). Optogenetic LGN activation thus elicited mostly tonic spikes, consistent with reports of a moderate propensity for bursting in tree shrew LGN (*Wei et al., 2011*). In order to assess to what degree LGN neurons were phase-locked to the optogenetic stimulation, we calculated phase-locking vector strength (see 'Materials and methods'). We found 12 LGN neurons in two animals that were significantly phase-locked (Rayleigh test p<0.05) at least at one of the flicker frequencies and across this population the magnitude of phase locking did not differ significantly over the four frequencies tested (*Figure 2B*). Tree shrew LGN layers are specialized for ON- and OFF-type visual inputs (see *Figure 2—figure supplement 1*; *Conway and*

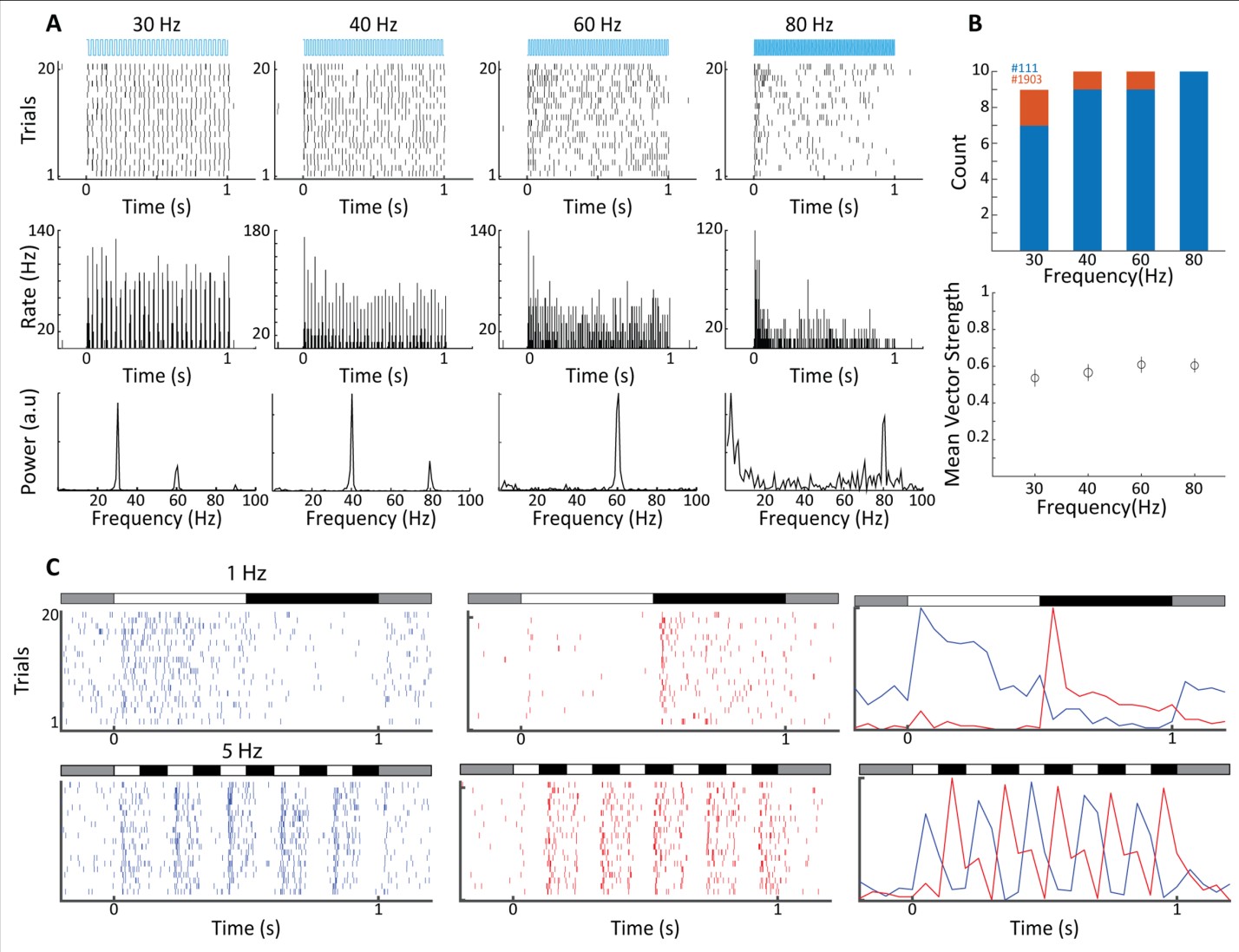

**Figure 2.** Electrophysiological validation of functional ChR2 expression in lateral geniculate nucleus (LGN) CaMKIIα neurons. (**A**) Raster plots (top) show 20 trials of responses of a single LGN neuron in animal #111 to different frequencies of 473 nm blue laser activation. Each vertical bar represents a single spike. Peri stimulus time histograms (PSTHs) (5 ms bin size) and Fast Fourier Transforms (FFTs) of the same spike trains are shown at center and bottom, respectively. For this, cell phase locking disappeared at the highest frequency, 80 Hz. (**B**) Top shows a histogram of the number of significantly phase-locked neurons, as determined by vector strength, at the four flicker frequencies for two animals (#111 blue, #1903 orange). Bottom is the mean vector strength for the phase-locked neurons at the four flicker frequencies; error bars represent SEM. (**C**) Raster plots (left and center) and PSTHs (right) reveal both ON- and OFF-type visual responses at sites nearby to laser stimulation for both 1 Hz top and 5 Hz bottom, stimulation frequencies. Bars at the top of each plot indicate the contrast condition, with gray being the background illumination.

The online version of this article includes the following figure supplement(s) for figure 2:

**Figure supplement 1.** Tree shrew lateral geniculate nucleus (LGN), functional organization.

*Schiller, 1983*), responding to onset of bright and dark targets with respect to background illumination, respectively. Local recordings of LGN neural activity at the site of light stimulation can thus be used to functionally identify the stimulated layer, as illustrated for 1 Hz and 5 Hz full-field bright visual stimulation (*Figure 2C*). Together, these results suggest robust expression of ChR2 in thalamocortical relay cells across tree shrew LGN layers under the CamKIIα promotor.

## Detection of a phosphene-like visual stimulus

To examine whether animals could detect LGN optogenetic activation, we first trained them on a visual detection task, where they had to enter a nose poke on a transparent horizontal platform at

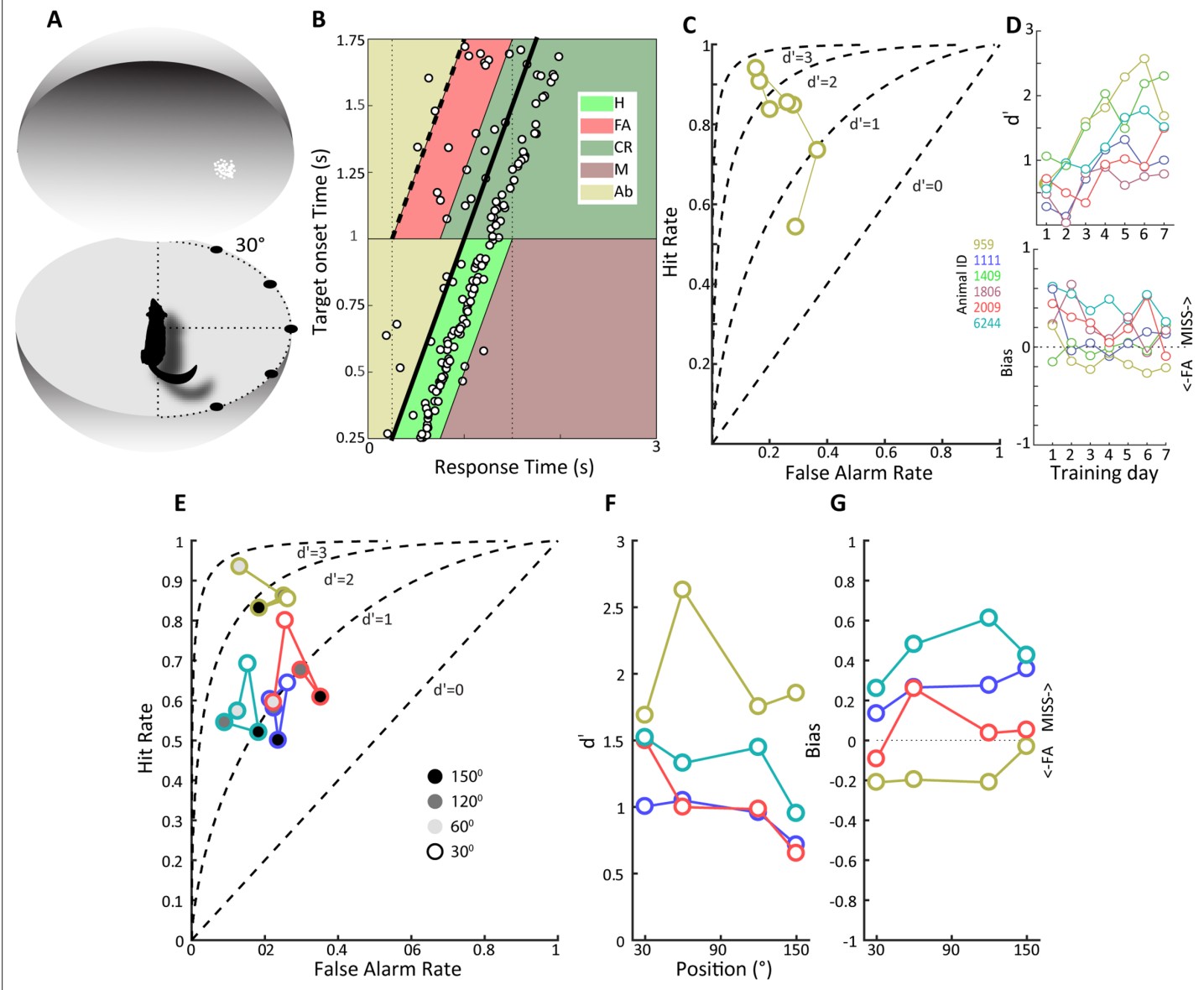

**Figure 3.** Detection of a visual moving dot stimulus. (**A**) Illustration of the experimental setup. Animals were trained to poke their nose into a response port at the center of a sphere and remain there until the appearance of a 2⁰ patch of moving white dots appeared in the field of view. (**B**) Plotted is the behavior of an animal, #959, that has acquired the task when phosphene stimuli were presented at the center of the field of view. Target onset is indicated by the thick, oblique black line. The light green trapezoid depicts the area during which responses were considered hits, and the orange trapezoid depicts the area of false alarms. H, hit; FA, false alarm; CR, correct rejection; M, miss; Ab, abort. (**C**) False alarm vs. hit rates are plotted over training days for the same animal as in (**B**). Note that as learning progresses the hit rate increases as the false alarm rate declines. (**D**) At top, d' values over training days are plotted for all animals, showing increased sensitivity over training days. Different colors denote different animals, and the color scheme is maintained throughout subsequent figures to facilitate comparison of individual animal performance on different tasks. (Bottom) Bias values for the same animals over training days. Note that bias values below zero indicate more false alarms and those above zero indicate more misses. (**E**) False alarm vs. hit rates are plotted for all animals for phosphene like moving dot visual stimuli presented at different positions relative to the center of the field of view, indicated by grayscale fill. (**F**) d' values as a function of stimulus position for the four animals tested. (**G**) Bias values for the four animals at the different stimulus locations.

The online version of this article includes the following figure supplement(s) for figure 3:

**Figure supplement 1.** Reaction times and d' over training days for visual detection experiment.

the center of a 70-cm-diameter opaque sphere and respond to the onset of a visual stimulus by withdrawing from the nose poke (see 'Materials and methods,' *Figure 3A*). Animals were thus in a defined position before and during visual target presentation, allowing us to assess their visual capacities in different parts of the visual field without the requirement of head fixation. We used a 2° visual stimulus composed of a cloud of moving white dots to facilitate the subsequent transition to the optogenetic stimulation detection task. Although artificial visual percepts can take on a variety of forms, they correspond better to moving clouds of dots rather than oriented bars or grating stimuli (*Fernández et al., 2021*). Behavioral data for an example tree shrew is shown in *Figure 3B* for a daily session consisting of 200 trials. It is evident that the animals' response times are narrowly distributed and largely follow the target onset time marked by the thick oblique line with a median response time of 290 ms and an overall correct performance of 76%. Our task did not contain any signal absent trials to avoid compromising animals' motivation for task performance, but we nevertheless used signal detection theory to estimate sensitivity and bias (see 'Materials and methods'). Briefly, trials where animals responded within 500 ms of target onset were designated as 'hits,' whereas responses made during an identical time window where no targets occurred were assigned as 'false alarms,' with corresponding 'miss' and 'correct rejection' assignments.

The example dataset in *Figure 3B* yields a d' sensitivity value of 2.6. The hit rate, false alarm rate, and d' values during the learning phase for this animal are shown in *Figure 3C*, illustrating increasing hit rates as well as decreasing false alarm rates during the 7-day learning period. Group data for the six tree shrews participating in this experiment is shown in *Figure 3D*, illustrating that all animals showed improved d' sensitivity over the course of learning, reaching d' values between 1 and 2.5 at the end of training. The d' sensitivity over the last four training days was significantly different from zero in all animals (*t*-test, p<0.01), confirming that all animals had acquired the visual detection task. Details of individual performance and reaction time as a function of training day can be found in *Figure 3—figure supplement 1*. The bias for these same behavioral sessions reveals considerable individual variation as well as a trend for reduced bias with training. Positive and negative bias values signal preponderance of 'miss' and 'false alarm' type errors, respectively. Interestingly, animals with similar and high d' values of around 2.0, that is, tree shrews 959 and 1409, exhibited opposite bias values, suggesting individual differences in strategy and decision threshold.

## The effect of eccentricity on visual stimulus detection

Once animals had acquired the detection task when visual targets were presented at visual field center, we proceeded to study how detection performance generalized to target presentations at other visual field locations in a subset of animals. As tree shrews have a wide visual field, we investigated eccentricities of up to 150° in four tree shrews, studying a single eccentricity in a single behavioral session. All animals readily generalized detection performance up to 120° (*Figure 3E and F*), with performance declining at 150°, the highest eccentricity tested, in all except one animal (959), which detected these peripheral moving dot stimuli without apparent problems. Note that slight differences in head position, estimated at ± 15°, cannot be ruled out in the nose poke task, so eccentricity values are not exact as they would be with head fixation. Eccentricity effects were studied in only a single session per eccentricity value; however, all tree shrews generalized significantly from the center position to eccentric stimulus presentation (*t*-tests comparing d' at four studied eccentricities against zero, p<0.01). Bias tended to increase with eccentricity (*Figure 3G*) as animals increasingly missed targets. Taken together, behavioral data on the visual detection task indicates that tree shrews can acquire this task and generalize across the visual field in a freely moving unrestrained behavioral setting, achieving d' values of up to 2.5 within a week of training.

## Detecting optogenetic activation of CaMKIIα neurons in the LGN

We next moved on to study if tree shrews pretrained on visual detection of a moving dot visual stimulus could also detect optogenetic activation in the LGN in the context of the same nose poke task. We initially performed some preliminary tests for detection of optogenetic activation using several frequencies (4, 10, 20, and 50 Hz) and amplitudes (3, 6.5 and 10 mW) in five tree shrews. We used a single combination of frequency and amplitude in each session, presented in randomized order using a wireless, subdermal radiofrequency-powered stimulation probe (*Shin et al., 2017*). An example of behavioral performance for a tree shrew implanted in the LGN (animal 1111) during a single session is

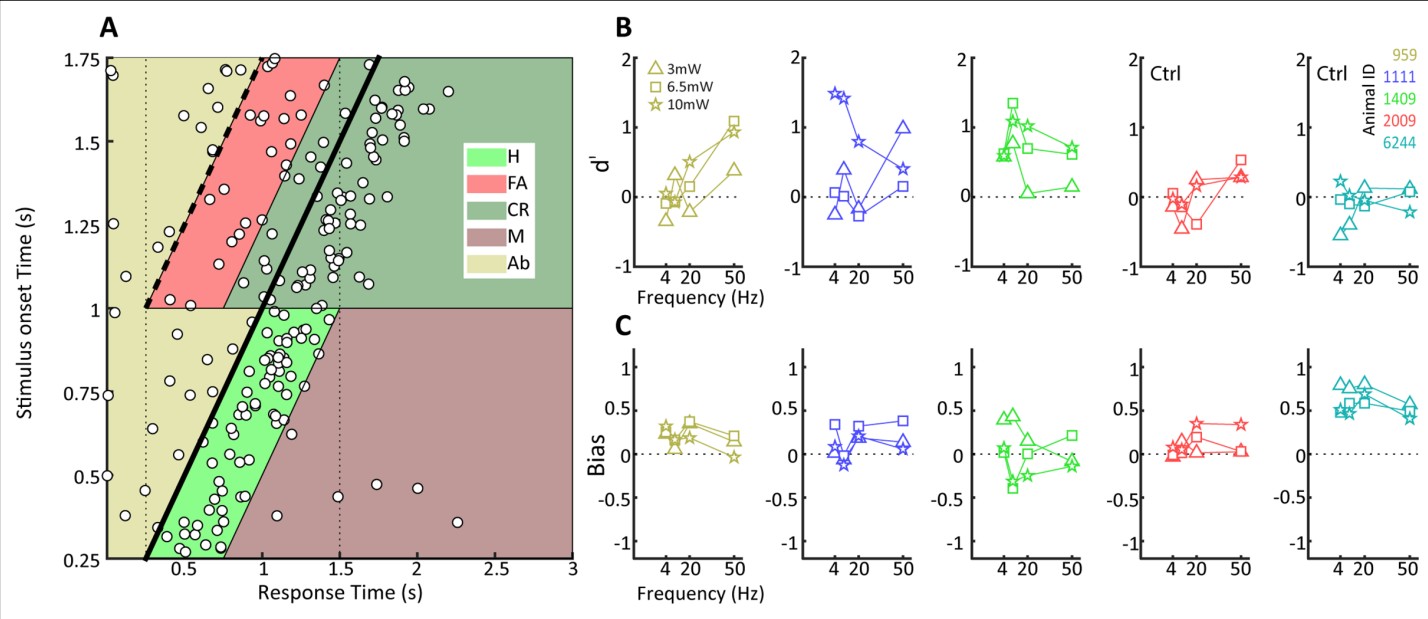

**Figure 4.** Behavioral detection of optogenetic activation of lateral geniculate nucleus (LGN) CaMKIIα neurons. (**A**) As *Figure 2B*, but for an animal, #1111, detecting blue LED activation of CaMKIIα neurons in the LGN. Note that while the animal can clearly perform the task, detection performance is more variable than for the visual stimulus. (**B**) d' values at the three LED intensities used are plotted for each animal at four stimulation frequencies (4, 10, 20, and 40 Hz). Ctrl indicates control animals. (**C**) Same as (**B**) but for bias scores.

The online version of this article includes the following figure supplement(s) for figure 4:

**Figure supplement 1.** Maximum d' in visual vs. optogenetic detection.

**Figure supplement 2.** Sequence of behavioral training.

shown in *Figure 4A*, corresponding to a behavioral performance of 63% correct trials with a median response time of 260 ms and a somewhat broader distribution compared to the visual detection task. Using the same procedure as above, we used signal detection theory to compute sensitivity (d') and bias for the optogenetic stimulation trials. Results for three tree shrews with AAV2 injections and LED implanted in the LGN (tree shrew 959, 1111, and 1409) and two tree shrews injected and implanted in control areas outside of the LGN (globus pallidus and zona incerta brain areas for tree shrews 2009, 6244) are shown in *Figure 4B and C* for sensitivity and bias. We note that LGN-implanted animals achieved d' sensitivity values around ≥1.0, whereas d' did not surpass 0.5 in control animals implanted outside the LGN (see *Figure 4—figure supplement 1* for a comparison of maximum d' values in visual and optogenetic detection tasks). This suggests that tree shrews can readily detect optogenetic activation of the visual thalamocortical pathway. Generally, higher amplitudes of stimulation tended to be detected most readily, as would be expected. In terms of light stimulation frequency, two animals (1111, 1409) appeared to respond best to 10 Hz stimulation while for one animal (959) the preferred frequency was 50 Hz. While it is possible that these results might be related to individual animal strategy toward interpreting the optogenetic stimulation, as has been reported also for humans in work using electrical stimulation (*Erickson-Davis and Korzybska, 2021*), they may well result from slight differences in spatial placement of the optic fiber relative to the transfected LGN neurons that can lead to hyperstimulation and reduced responsiveness (*Kittelmann et al., 2013*). Importantly, tree shrews showed evidence for rapid generalization from the visual to the optogenetic stimulation. This is demonstrated in *Figure 4—figure supplement 2A*, where the d' values are shown for sessions with different amplitude and frequency combinations in serial order of occurrence; the same order was used in each animal. *Figure 4—figure supplement 2B* shows the performance during the first day of optogenetic LGN activation (10 Hz frequency at 10 mW) for the three tree shrews, illustrating best generalization from the visual detection task (tree shrew 1111) already early during the session, as well as a trend for within-session behavioral improvement particularly in the other two tree shrews (animals 959 and 1409). To examine variability and reproducibility of optogenetic stimulus detection across sessions in individual tree shrews, we performed multiple sessions in each of the three animals

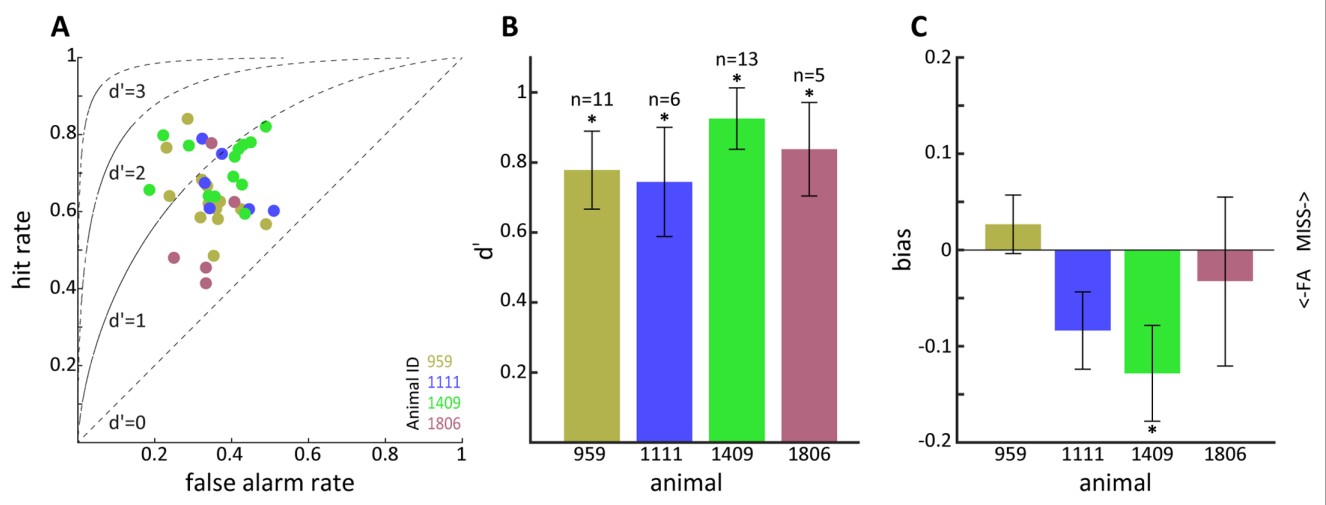

**Figure 5.** Sensitivity to optogenetic activation of lateral geniculate nucleus (LGN) CaMKIIα neurons. (**A**) Hit vs. false alarm rates for multiple sessions. (**B**) Mean d' calculated from (**A**) for each animal; error bars reflect SEM and * indicates p<0.05. (**C**) Same as (**B**) but for bias scores; note that animal 1409 showed significant bias for false alarms.

using the preferred stimulation frequency for each animal at 10 mW amplitude, as well as an additional animal (1806) for which we used 10 Hz stimulation. The results, shown in *Figure 5*, highlight that tree shrews reliably detected optogenetic stimulation as evidenced by significant d' values (two-way ANOVA main effect, p<0.01). A significant effect of bias between animals is apparent (one-way ANOVA, p<0.05), suggestive of differences in strategy during the optogenetic detection task, with tree shrew 1409 showing significantly negative bias. These findings document the reproducibility of tree shrew behavioral detection of optogenetic LGN activation.

## Electrophysiological correlates of LGN CaMKIIα ChR2 activation

We proceeded to study neural circuit activity in the visual pathway triggered by LGN CaMKIIα-ChR2 activation, simultaneously targeting both the LGN at a location of the transfected neurons as well as the primary visual cortex (V1) while aiming for maximum spatial overlap in LGN and V1 receptive fields. To achieve this, we first determined the receptive field location of the LGN at the site of virus injection, and then performed a V1 electrode penetration at a position that corresponded to the

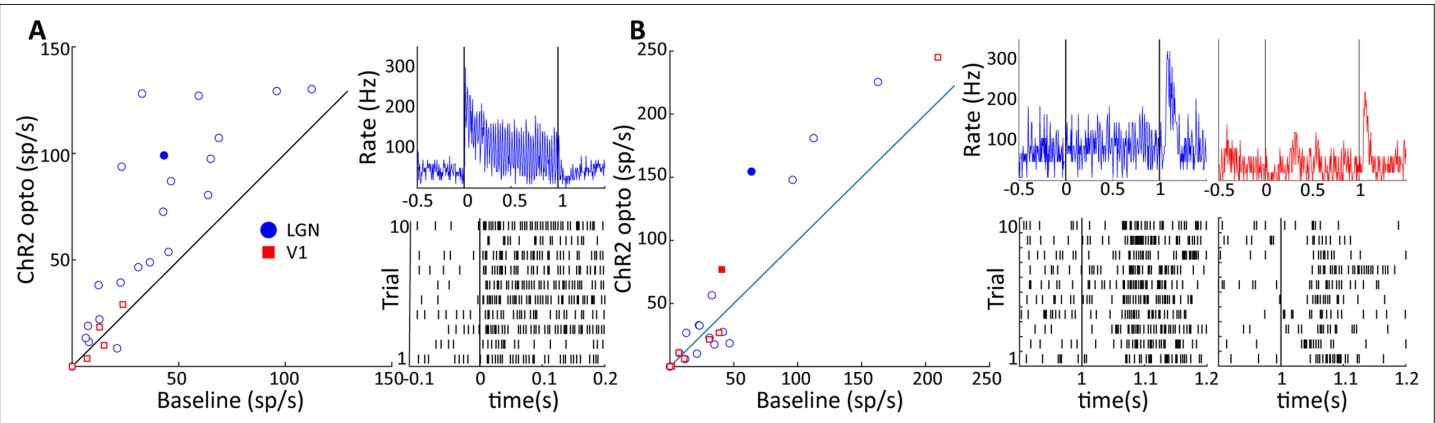

**Figure 6.** Two response motifs in lateral geniculate nucleus (LGN) and V1 following optogenetic excitation of LGN CaMKIIα neurons. (**A**) and (**B**) show, respectively, sustained, low latency-type responses and those showing only 'OFF'-type responses in LGN (blue) and V1 (red). (**A**, left) Scatter plot comparing the firing rate of significantly modulated neurons in LGN and V1 during a baseline period and during 40 Hz optogenetic activation of the LGN. (**A**, right) PSTH top and raster plot bottom of an example LGN neuron, filled blue circle in (**A**), in response to 40 Hz blue laser activation of the LGN. Each vertical line in the raster plot corresponds to the time of one spike. (**B**) Same as (**A**), but for V1 and LGN neurons showing only 'OFF'-type responses. Example neurons shown in right panels correspond to the filled blue circle, and filled red square for LGN and V1, respectively.

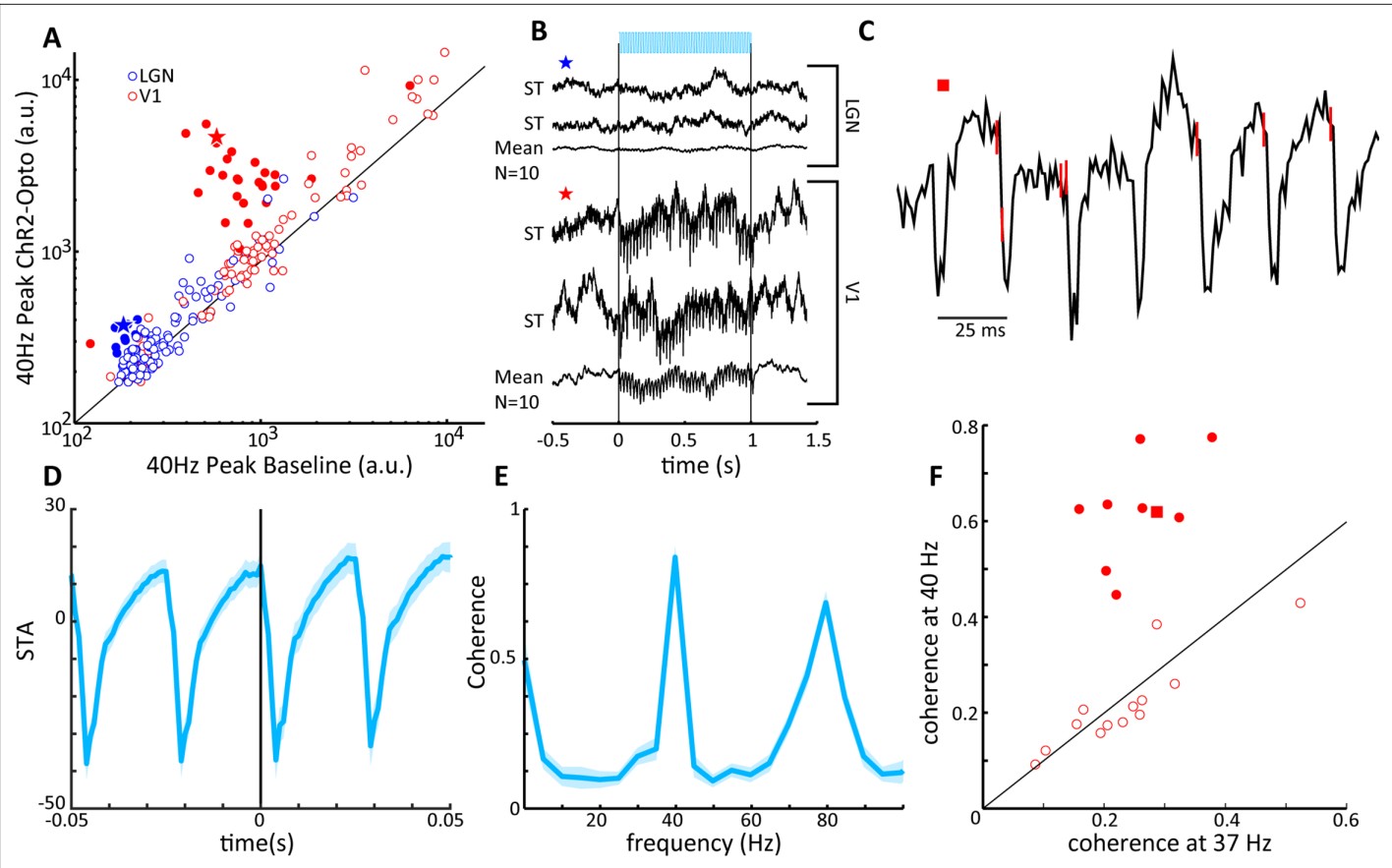

**Figure 7.** V1 local field potentials (LFPs) show temporal coherence with laser-induced spiking in lateral geniculate nucleus (LGN). (**A**) Scatter plot shows the magnitude of the spectral peak at the frequency of LGN laser stimulation (40 Hz) for baseline and LGN stimulation conditions, LGN (blue), V1 (red). Filled circles represent LFP locations showing a significant increase in power at the stimulation frequency. (**B**) Two single-trial traces (ST) and the mean of 10 trials of LFPs in LGN (top) and V1 (bottom) during 40 Hz laser stimulation of the LGN. Example traces correspond to the star symbols in (**A**). (**C**) LGN spikes (red vertical lines) are superimposed upon the V1 LFP during a single trial of 40 Hz laser activation of LGN. Note that spiking in the LGN is quickly followed by a large depth negative deflection in the V1 LFP. (**D**) Spike-triggered average of the V1 LFP with spiking in the LGN. (**E**) Spike field coherence between spiking activity in the LGN and the V1 LFP showing high coherence at the stimulation frequency, 40 Hz. (**F**) Coherence at the stimulation frequency (40 Hz) is compared to coherence at a shoulder frequency, 37 Hz, for all V1 LFPs showing significant modulations in (**A**). Filled symbols reflect significant differences, and the square symbol represents the example in (**C**).

retinotopic position of LGN activation. We then analyzed neural spiking in terms of multi-unit activity (MUA) in LGN and V1, focusing on the sustained response during the 40 Hz 50% duty cycle optogenetic activation period. We observed significant activation in 22 out of 111 LGN MUA sites, but only in 5 of 111 V1 MUA sites (paired *t*-tests: p<0.05, ***Figure 6A***). Many of the activated LGN sites showed a robust, short latency response motif to stimulation (see example unit in ***Figure 6A***), while our recordings contained only relatively weakly modulated V1 MUA sites at short latency. We observed that the MUA at some LGN and V1 sites exhibited offset responses following the termination of the light stimulation (LGN: n = 16, V1: n = 8, paired *t*-tests: p<0.05, ***Figure 6B***). Robust offset responses were evident in both LGN and V1 (see ***Figure 6B***) and had response latencies of around 50 ms following the cessation of stimulation in both areas, consistent with a neural circuit origin of these effects involving recurrent processing and inhibitory modulations. The unit data highlights that LGN optogenetic stimulation robustly drives local units, and while it was more difficult to isolate V1 units directly modulated by optogenetic stimulation, possibly due to incomplete overlap of the stimulated thalamocortical recipient zone with the receptive field of the V1 MUA activity.

We next examined the effect of LGN CamKIIα-ChR2 light stimulation on local field potentials (LFPs). While the success rate of finding V1 units whose activity was modulated by LGN activation was low (see above), V1 LFPs were strongly modulated by stimulation in the LGN, as is shown by comparing

the peak in the Fourier spectrum during the 40 Hz stimulation condition to the pre-stimulation baseline period (**Figure 7A**). Indeed, 29/111 V1 sites showed significant 40 Hz entrainment (paired *t*-test: p<0.01), and traces from an example site illustrate high-amplitude 40 Hz oscillations during stimulation on two single trials as well as the average across repetitions (**Figure 7B**). These V1 sites were thus clearly within the thalamocortical recipient zone with significant receptive field overlap between LGN and V1 sites. The thalamic LFPs were by comparison rather insensitive to light stimulation, with only 11/111 sites showing significant modulations (paired *t*-test: p<0.01; $\chi^2$-test comparing LGN and V1; p<0.01, see **Figure 7A**). This is also evident in single trial and averaged LGN LFP traces (**Figure 7B**), where, although significant, the oscillations are quite modest in amplitude, similar to reports in macaque (**Bastos et al., 2014**). Endogenously generated LFPs during the baseline period in LGN were much lower in magnitude than in V1 (for 40 Hz unpaired *t*-tests: p<<0.001) since unlike V1 the LGN does not possess systematically aligned dendritic structures that underlie LFP activations (**Rainer, 2019**). Since our data was collected simultaneously using similar electrodes of comparable impedance, we can here exclude technical factors as sources for this difference between LGN and V1 LFPs.

In order to further assess the entrainment of the V1 LFP to LGN optogenetic activation, we next examined the spike-triggered average (STA) of LGN MUA with the V1 LFP. An example trace of the V1 LFP from a single trial during LGN activation is shown in **Figure 7C**, with LGN spike times denoted by red vertical lines on the trace. From this example, it is apparent that LGN spikes were shortly followed by sharp negative deflections in the cortical LFP consistent with triggering activation of thalamocortical projections. This can be seen in the average STA, where a strong phase-locked relationship was present at the stimulation frequency of 40 Hz (see **Figure 7D**), which is recapitulated in a different measure of functional neural coupling, the spike field coherence (SFC) (**Figure 7E**), showing clear peaks at both the fundamental and the first harmonic of the 40 Hz stimulation frequency. Indeed, out of a population of 22 V1 sites that showed significant increases in 40 Hz power following optogenetic activation, 9 of these sites also exhibited significant coherence with LGN spiking at 40 Hz (see **Figure 7F**). Note that V1 sites lacking oscillatory entrainment are most likely explained by only partial receptive field overlap between LGN and V1 sites. We took the time of first negative peak of the STA as the response latency to the thalamic spikes, M = 3.8 ± 1.5 ms (n = 9), with no significant difference across stimulation frequencies (p>0.05, one-way ANOVA), which suggests that the LGN is capable of monosynaptically driving V1 oscillatory activity at least up to 80 Hz without significant degradation of the entrainment. These functional coupling results thus support the notion that CaMKIIα thalamocortical projections can activate V1 circuits.

For a limited set of recordings, we explored the impact of variations in light stimulation frequency on V1 LFPs while maintaining 50% duty cycle and thus keeping overall light illumination constant across

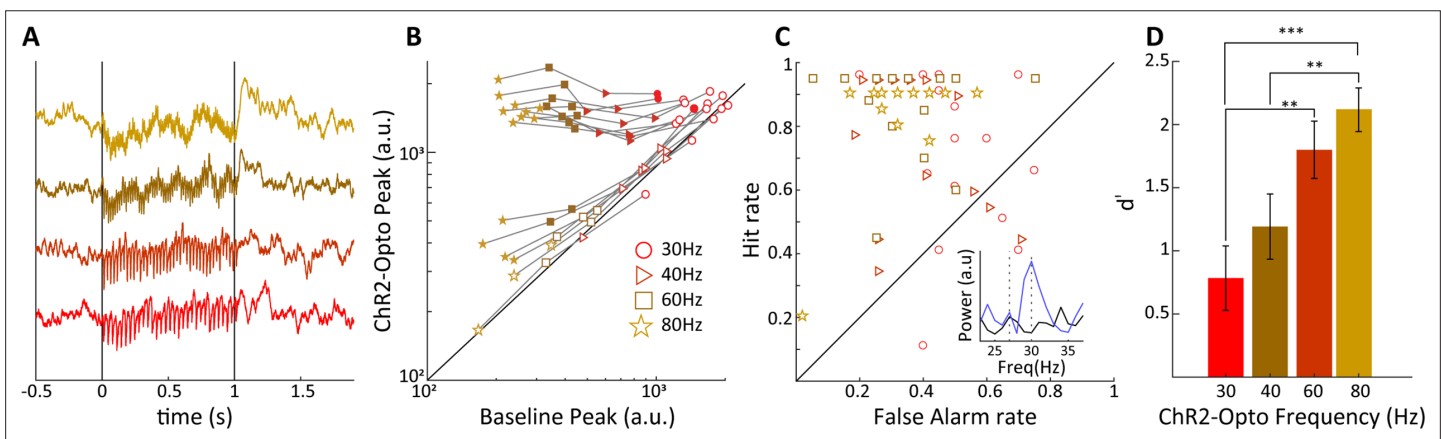

**Figure 8.** Sensitivity of V1 local field potentials (LFPs) are frequency dependent. (**A**) V1 LFPs averaged over 20 trials at the four stimulation frequencies, 30, 40, 60, and 80 Hz, from bottom to top. (**B**) The magnitude of the spectral peak at the four stimulation frequencies during baseline vs. lateral geniculate nucleus (LGN) laser stimulation conditions. Note that the filled symbols indicate a significant effect of optogenetic stimulation. (**C**) Hit vs. false alarm rates, see text, for all V1 recording sites are plotted for each frequency of laser stimulation. Inset shows an example FFT during baseline (black) and laser stimulation (blue) during 30 Hz laser stimulation of the LGN. (**D**) Average V1 LFP sensitivity, d', at the four laser stimulation frequencies. Note the monotonic increase in sensitivity with frequency. Error bars reflect SEM.

conditions. In *Figure 8A* we show trial-averaged evoked LFPs for 30, 40, 60 and 80 Hz optogenetic stimulation, revealing notable frequency-specific modulations whose amplitude is inversely correlated with frequency, as is indeed generally the case for brain potentials. In *Figure 8B*, the evoked Fourier spectral peak is compared to the control condition, for a set of 16 sites. At 30 Hz frequency, only 3/16 sites were significantly modulated in spite of the high overall amplitude of the induced modulations, likely because endogenously generated activations were already relatively prominent at this frequency. With increasing frequency, we found that a number of sites exhibited pronounced and significantly elevated spectral peaks at the stimulation frequency, consistent with direct thalamocortical drive reaching the V1 site (e.g. nine sites at 80 Hz stimulation, paired *t*-test: p<0.01). At 80 Hz, four additional sites showed more moderate, but still significant evoked spectral peaks, likely due to the V1 site being somewhat distant to the thalamocortical recipient zone. We used the Fourier results to compute hit and false alarm rates assessing the capacity of V1 sites to detect optogenetic stimulation by comparing the spectral peak at stimulation frequency ($f_{stim}$) to a control frequency at $f_{stim}$-3 Hz (see *Figure 8C*, 'Materials and mmethods'). We found higher frequency stimulation tended to result in enhanced hit rates and reduced false alarm rates. Converting these data to d' values, we found that indeed the sensitivity of V1 sites increased significantly with light stimulation frequency, reaching values of d' ≈ 2.0 at 80 Hz. Note that for the frequencies used in the behavioral studies up to 50 Hz, the sensitivity of V1 LFPs appears to generally correspond to the sensitivity of the tree shrews at the behavioral level, that is, values in the range of 1–1.5. A caveat is that V1 LFPs were recorded during general anesthesia, such that in the awake state additional endogenous activations might possibly reduce or enhance the d' sensitivity. Previous data suggests that membrane depolarization triggered by ChR2 light stimulation are attenuated with increasing stimulation frequencies upward of 10 Hz (*Erofeev et al., 2019*; *Tchumatchenko et al., 2013*). While robust transmembrane currents remain at 100 Hz light stimulation, these are attenuated by a factor of around 70-fold relative to 10 Hz stimulation compared to a 10-fold attenuation for 50 Hz light stimulation (*Malyshev et al., 2015*), and attenuation can be dependent on cell type and choice of AAV delivery system (*Jackman et al., 2014*). The entrainment observed in this study nevertheless indicates that CamKIIα-ChR2 expression in LGN neurons permits efficient activation of thalamocortical projections up to at least 80 Hz stimulation frequency, and that V1 LFPs are sufficiently sensitive to permit accurate single-trial detection of LGN optogenetic stimulation.

## Discussion

We demonstrate here that optogenetic activation of the visual thalamic LGN triggers local neuronal circuit activity as well as an associated artificial visual percept that can be readily detected by the animals. A previous report has already provided proof of principle that electrical LGN stimulation can elicit artificial visual percepts in macaque monkey (*Pezaris and Reid, 2007*). This previous study used a visually guided saccade task instead of the nose-poke detection paradigm in our study, but essentially both tasks permit the characterization of behavioral responses to natural and artificial visual percepts across the visual field. As did the macaques in the previous study, tree shrews quickly generalized from the visual to the optogenetic task, suggestive of a visual nature of the evoked artificial percept. The overall task performance we observed for optogenetic stimulation remained somewhat lower than the values observed during visual stimulation, suggesting that artificially induced percepts were overall less readily detectable than visually induced percepts at the high contrast employed in this study. Since V1 responses scale with stimulus contrast (*Bhattacharyya et al., 2013*), artificial stimulation detectability thus matches visual detectability at low to intermediate contrast values. We observed some interindividual variability between tree shrews in terms of initial acquisition of the optogenetic stimulation detection task, as well as in terms of behavioral strategy after learning as demonstrated by differences in bias. Indeed, it has been shown that for macaques, extensive training on detection of electrical microstimulation in V1 can be detrimental for detection of visual stimuli (*Ni and Maunsell, 2010*), consistent with an important role of behavioral strategy during optogenetic stimulus processing. We found some variability in terms of optimal stimulation frequency between animals, probably related to variations in factors such as virus expression and LED probe placement. Indeed, computational modeling work relating to V1 suggests that the spatiotemporal configuration of light delivery and virus transfected neural elements including dendrites plays a crucial role in how optogenetic activation is translated into cortical activity and ultimately perception (*Antolik et al.,*

*2021*). Further work extending computational modeling of electrical activations (*Jawwad et al., 2017*) to optogenetics in the LGN is necessary toward ideally refining the models to integrate available experimental data. Our present results suggest that optogenetic LGN stimulation, up to the light intensities tested here, produces robust artificial percepts that tree shrews can reliably detect in a suitable behavioral paradigm. Increasing stimulation intensity further may be one option to further enhance performance as long as this does not cause tissue damage, nonspecific activation due to heating or undesirable loss of spatial selectivity (*Stujenske et al., 2015*). Note that overly strong LGN stimulation is in any case undesirable as it has been shown to deactivate visual brain areas downstream of V1 for electrical stimulation (*Logothetis et al., 2010*), which would certainly compromise goal-directed behaviors based on artificial visual percepts.

There are other more principled factors that might limit the detectability of optogenetic or electrical stimulation since these methods will activate populations of nearby neurons that might not be coactivated during natural vision, triggering comparatively diffuse and weak phosphene percepts and complicating behavioral decision-making based upon artificial stimulation. For example, since visually responsive ON- and OFF pathways are intermingled in LGN and layer 4 of V1 in most species including primates, artificial stimulation necessarily co-activates both contrast channels, which never occurs in natural vision since a part of the visual field can be bright or dark but not both at the same time (*Beyeler et al., 2017*; *Nguyen et al., 2016*). Such coactivation will thus produce cortical activation patterns not encountered during natural vision, potentially explaining the perceptual performance decrement for artificial vision. In the tree shrew LGN, ON- and OFF neurons tend to be segregated in separate layers, with layers 1 and 2 containing ON cells, layers 4–5 containing OFF cells, layer 3 containing a mixed population, and layer 6 containing ON/OFF cells with an inhibitory surround *Conway and Schiller, 1983*; note that we have adopted the conventional labeling of the LGN layers as 1–6 from medial to lateral, whereas in Conway this ordering is reversed. Since our viral injections spanned the LGN layers, we did not specifically target ON- or OFF pathways as our focus was obtaining proof of feasibility related to the perception of LGN activation. Future experiments, targeting, for example, exclusively ON layers 4–5 of tree shrew LGN, could, however, address the issue of specific contribution of ON/OFF activation in artificial vision. We predict that such stimulation might trigger artificial percepts to which animals exhibit elevated d' sensitivity values more similar to those for natural visual stimuli, and that detection performance under these conditions will be more sensitive to a visual masking stimulus containing dark structure rather than bright structure on a gray background. The functional anatomy of the tree shrew visual pathway permits a systematic investigation of this issue, providing an intermediate step before the issue can be addressed in macaques where LGN layers are not segregated into ON and OFF pathways (*Schiller and Malpeli, 1978*), in a translational neuroscience context toward eventual human clinical application development.

Our simultaneous recordings in LGN and V1 provide a number of interesting insights into the circuit activations triggered by optogenetic LGN stimulation. We found neural spiking activations linked to optogenetic stimulation particularly in LGN and for a small number of neurons also in V1. While most neurons were activated by the onset of LGN optogenetic stimulation (ON-type response), a minority of neurons interestingly exhibited significant OFF-type responses and were thus activated when the optogenetic stimulation ceased. OFF-type responses are commonly observed to visual stimulation, for example, in the macaque (*Bair et al., 2002*), where they tend to occur with shorter latencies than ON-type responses in both LGN and V1. In our study, we found the opposite effect for optogenetic stimulation in that the OFF responses were delayed compared to ON responses with longer latencies of about 50 ms in both LGN and V1. Bair and colleagues argue that due to the shorter latency and lower stimulus-linked variability, OFF responses might in fact serve as a reference signal for temporal segregation of information by the visual system. Altered OFF responses following optogenetic stimulation might thus complicate perception and behavioral stimulus interpretation in artificial vision. The apparently inverted dynamics between natural and artificial visual stimulation is therefore of interest and will need to be investigated further, for example, in terms of dependence of activated cell type, as it reveals a divergence in dynamic neural circuit activation between these two stimulation conditions.

In order to further understand the impact of optogenetic activation of CaMKIIα LGN neurons on V1 activity we examined the STA and SFC of the V1 LFP using optogenetically triggered LGN spiking as the reference. We found that laser-induced spikes in the LGN were reliably followed by depth negative potentials in V1 at a latency of ≈3 ms, which is similar to the latency of the V1 excitatory

postsynaptic potential (EPSP) triggered by monosynaptic input from LGN neurons reported in the cat (*Sedigh-Sarvestani et al., 2017*). As the LFP is thought to primarily reflect synchronized synaptic inputs, this indicates that activation of CaMKIIα LGN neurons is capable of modulating V1 responses at the earliest stages of cortical processing. Additionally, the V1 LFP was spectrally coherent with evoked LGN spiking across the gamma frequency range (30–80 Hz). As discussed above, oscillatory activity in the gamma band has been associated with a number of visual processes, and it is becoming increasingly clear that different frequencies of gamma activity are associated with different stimulus parameters. Here we were capable of tuning the frequency of V1 gamma activity through LGN opto-genetic activation, a potentially important step in the development of visual prosthetics. The relationship between LGN spiking and narrow band gamma activity in V1 has previously been shown to be coherent at both the level of V1 EPSC's, as well as the cortical LFP (*Saleem et al., 2017*; *Schneider et al., 2021*). Importantly this coherence is thought to be mitigated strictly by thalamocortical inputs, suggesting that a cortically driven neural prosthetic would be insufficient to generate thalamocortical coherence, which may be critical to the generation of organized visual percepts (*Ribary, 2005*).

We show that optogenetic stimulation frequency has a profound effect on the detectability of V1 LFP oscillations, such that V1 LFP sensitivity parameter d' is enhanced with increasing stimulation frequency. For example, 80 Hz stimulation triggers much more reliable neural circuit activation than 30 Hz stimulation, suggesting that based on this criterion alone higher stimulation frequencies might be preferable for use in artificial vision. Note that the relationship between LFP gamma oscillations and spiking activity is complex (*Ray and Maunsell, 2011*; *Ray and Maunsell, 2010*), but for the purposes of artificial vision we consider that reliable gamma oscillations will generally be accompanied by robust neural spiking activity in V1 circuits. Insight into the potential nature of the evoked percept comes from a recent study that has examined how drift rate of a visually presented stimulus impacts V1 gamma oscillations (*Salelkar et al., 2018*). The authors present a dissociation between low (under 40 Hz), narrow-band (50–80 Hz), and high gamma (above 140 Hz) in terms of which drift rate elicited highest oscillatory activity. While low gamma responded optimally to high drift rates, narrow-band gamma had a preference for low drift rates and high gamma tended to prefer intermediate drift rates. Importantly, the most reliable oscillations were evoked in the narrow-band gamma range, as highlighted by the tuning fit quality parameter. Translated to our study, this suggests that 30 Hz optogenetic stimulation, falling in the low gamma range, might trigger fast-drifting phosphene-like percepts, whereas 60–80 Hz stimulation should by contrast trigger intermediate drift-speed phosphenes. Related to this, distinct sensitivity in different gamma bands to spatial frequency (SF) of visual stimuli has also been demonstrated, such that low gamma (25–40 Hz) exhibits preference for high SF, medium gamma (45–60 Hz) prefers medium SF, and high gamma (65–90 Hz) responds best to low SF. Note that while low and medium gamma appear to be generated in V1, high gamma appears to have its origin in the LGN (*Han et al., 2021*), suggesting that optogenetic stimulation upward of 65 Hz might trigger visual circuit activations more closely resembling endogenous activations caused by visual stimulation. Future studies combining optogenetic and natural visual stimulation are necessary to test these hypotheses and offer an avenue for detailed characterization and optimization of artificially induced percepts.

In summary, our findings reinforce the idea that the LGN might be a particularly promising target for artificial vision (*Nguyen et al., 2016*). Our findings suggest that the tree shrew, as a small diurnal highly visual mammalian species, appears well suited for translational animal work in the area of visual prosthetics before the validated approaches are tested in macaques and eventually humans. We have started here with optogenetics using the CamKIIα promotor and shown that tree shrews can readily detect light stimulation delivered within the LGN. Expanding the scope to additional promotors, for example, targeting other classes of thalamocortical relay cells, as well as multiple light sources that allow dynamic, sequential activation of neural circuits (*Beauchamp et al., 2020*) are highly promising avenues toward a future generation of prosthetic devices. For example, intermittent optogenetic inhibition of LGN inhibitory cells could provide an efficient way to trigger coordinated activation in nearby relay neurons. Our findings show robust expression of light-sensitive opsins in layer 2/3 as well as layer 4 of cortex, opening the possibility of illuminating the thalamocortical axon terminals by light arrays overlying the cortical surface. Such devices can be manufactured as flexible implants, which could eventually deliver stimulation in a significantly less invasive manner compared to implanted electrodes, highlighting a potential advantage of the LGN as a target for artificial vision.

## Materials and methods

The local ethical committee on animal experimentation (canton of Fribourg) approved all experimental procedures (license number: 33056).

### Subjects

Ten male and female tree shrews (*Tupaia belangeri*) weighing 235 ± 10 g (0.6–2 years old) participated in the experiments, six in the behavioral studies and four in the neurophysiological studies. They were given ad libitum access to food and water and housed in temperature-controlled rooms (ambient temperature: 26 ± 1°C, air humidity: 60 ± 5%) under a 13/11 light/dark cycle with gradual illumination transitions at the beginning and end of the light period.

### Behavioral training

Tree shrews were first shaped to enter a nose poke mounted on a transparent horizontal platform at the center of a 70-cm-diameter opaque sphere for mango juice reward using continuous reinforcement in daily sessions lasting 30 min in the absence of food or water restriction. The mango juice was delivered at a fluid port located on the edge of the platform behind the nose poke and adjacent to the sphere wall.

Visual stimuli were projected onto the sphere using an Optoma GT-1080 E projector with a 1080p resolution. All visual stimuli were presented on a uniform gray background, 2.5 cd/m$^2$ at a distance of 35 cm. For the visual detection task, animals were trained to wait in the nose poke until a 10° × 10°, 45 cd/m$^2$ luminance square was projected onto the globe directly ahead of the animal. The target onset times were gradually increased from fixed 250 ms to fixed 250 ms + randomized (0–1500 ms) duration, while concurrently the response window for obtaining a reward was decreased from 1200 ms to 500 ms. Once animals achieved good performance on this task, a cloud of 50 bright dots 0.25° in diameter, moving at 0.3°/s in random directions with a total luminance of 39.4 cd/m$^2$, were presented. The moving dot stimulus had a random onset time between 250 and 1750 ms. Animals participated in seven training sessions with the visual stimulus in a fixed position (15° elevation, 30° right of center) (see *Figure 3A–D*). For the visual field generalization task, the moving dot stimulus was presented in consecutive sessions at 30, 60, 120, and 150° eccentricity. Behavioral sessions generally lasted 200 trials, taking approximately 30–60 min.

For the optogenetic detection task, parameters were identical to the visual detection task above, except that the visual stimulation was replaced by laser activation at 50% duty cycle with frequencies of 4, 10, 20, and 50 Hz with a stimulation period of 500 ms at amplitudes of 3, 6.5, or 10 mW. In *Table 1*, we summarize which tree shrew participated in which task, with serial order from left to right corresponding to the sequence of task participation.

For the optogenetic stimulation with multiple frequencies and amplitudes, we used the same serial order of conditions in all participant animals (*Table 2*), and starting with a block of 10 W stimulation to facilitate similar transition from the visual to the optogenetic task in all animals. Thus, the differences in behavioral performance we observed are not attributable to the sequence of amplitude/frequency conditions. For the optogenetic detection experiment with fixed parameters, we used

**Table 1.** Summary of participation of tree shrews in behavioral tasks.

| Tree shrew | Visual detection | Spatial generalization | Optogenetic detection multiple parameters | Optogenetic detection multiple parameters control animals | Optogenetic detection fixed parameters |
|---|---|---|---|---|---|
| Figure panels | *Figure 3C and D* | *Figure 3E–G* | *Figure 4B* | *Figure 4B* | *Figure 5A–C* |
| 959 | x | x | X | | x |
| 1111 | x | x | X | | x |
| 1409 | x | | X | | x |
| 1806 | x | | | | x |
| 2009 | x | x | | x | |
| 6244 | x | x | | x | |

10 Hz stimulation in animals 1111, 1409, and 1806 and 50 Hz stimulation frequency for animal 959; always at amplitude of 10 mW, 50% duty cycle, and 500 ms duration.

## Behavioral data analysis

The use of signal detection theory (*Green and Swets, 1966*) for our detection task data requires estimation of hits, false alarms, misses, and correct rejections (see *Figure 3A*). We thus partitioned the trials into two halves depending on target onset time, each of 750 ms duration. For onset times from 250 to 1000 ms, responses that occurred after target onset and before the 500 ms response window were assigned as 'hits,' and responses occurring after the response window were assigned as 'misses.' We then constructed a time window identical to the 'hit' window for the second half of the trials with onset times 1000–1750 ms and used this to estimate 'false alarms.' In terms of timing, 'hit' and 'false alarm' windows are indistinguishable to the animal as these windows cover identical time periods, the only difference being that no target occurs in the 'false alarm' window. If animals do not respond during the 'false alarm' window, the trial is assigned as 'correct rejection.' This procedure permits the computation of sensitivity (d') and bias (b) parameters according to signal detection theory for response times generated in a simple detection task. Note that this procedure was used for analysis only, and animals were always rewarded only for nose poke withdrawal upon stimulus presentation. Responses occurring prior to the 'hit' and by extension 'false alarm' windows were considered as aborted trials. Rates were computed as follows: $Hr = H/\left(H + M + Abort_{250ms}^{1000ms}\right)$, $FAr = FA/\left(FA + C + Abort_{1000ms}^{1750ms}\right)$. Sensitivity: $d' = z\left(Hr\right) - z\left(FAr\right)$, Bias: $b = \frac{-1}{2}\left(z\left(Hr\right) + z\left(FAr\right)\right)$. Excellent behavioral performance on decision tasks is associated with high sensitivity and low bias.

## Surgical procedures

For both viral injections/LED stimulator implantation and neural recordings, general anesthesia was induced by alfaxalone (40 mg/kg, half dose in each leg, intramuscular) (*Gehrig and Moens, 2014*), and we administered atropine (0.08 mg/kg, intramuscular) to reduce secretions and Baytril (2.5%, 0.3 ml/kg, subcutaneously) to prevent infection. Anesthesia was maintained by isoflurane (0.5–3.5%) in 100% $O_2$ administered using a nose cone for viral injections or an endotracheal intubation for neural recordings. Here, after identifying the vocal cords, a PE50 polyethylene tubing extending 1 cm from an endotracheal tube (Original Perfusor type: IV-standard-PVC, 6 cm, 1.5 × 2.7 mm) was inserted into the trachea under visual inspection using video laryngoscopy (*Balzer et al., 2020*). We monitored exhaled $CO_2$ (Physiosuite, Kent Scientific) and ventilated animals at 100 bpm (small animal ventilator 683, Harvard Apparatus). Animals were placed in a stereotactic apparatus, skin was shaved, periosteum retracted, cranial bone cleaned (3% $H_2O_2$), and craniotomies were drilled to allow access for viral injections or neural recording. Postoperative analgesia was administered before the end of surgical intervention (buprenorphine, 0.05 mg/kg, subcutaneously). We injected the construct AAV2-CamKIIa-hChR2(H134R)-mCherry (1 µl, UNC Vector Core) unilaterally using a microsyringe (34 GA. beveled Needle, 10 µl NANOFIL syringe, WPI) into the left LGN (AP 3.1, ML 4.8, DV from 5 to 7 dependent on the neural response for 5 Hz full-field flickering stimulus to ensure accurate placement). Following viral injection, in five of the animals a wirelessly powered device containing the µ-LED at the tip end of a freely adjustable needle (NeuroLux, Chicago, IL) was lowered into the LGN, again under stereotactic guidance, and cemented in place with dental acrylic (Paladur). In an additional animal, 1806, a similar but novel device with a longer shank was inserted. This wireless device was fabricated following standard flexible electronics manufacturing. Each device contains an 11-mm-long, 0.4-mm-wide, and 0.1-mm-thick probe, which is reinforced with 50-µm thin tungsten needle to prevent mechanical buckling during implantation. The probes are equipped with blue µ-LEDs (470 nm, 220 µm [w] × 270 µm [l] × 200 µm [th]), each emitting 21 mW/mm². The devices were encapsulated with a bilayer of parylene-C (14 µm) and polymethyl siloxane (PDMS, ~50 µm at the probe) to create a robust biological fluid barrier (Parylene-C) and increase the mechanical compliance (PDMS).

**Table 2.** Serial order of frequency and amplitude values for experiment with multiple parameters.

| Amplitude (mW) | 10 | 10 | 10 | 10 | 3 | 3 | 3 | 3 | 6.5 | 6.5 | 6.5 | 6.5 |
|---|---|---|---|---|---|---|---|---|---|---|---|---|
| Frequency (Hz) | 10 | 20 | 4 | 50 | 10 | 20 | 4 | 50 | 10 | 20 | 4 | 50 |

After viral injections/LED stimulator implantation, the skin was sutured and animals were allowed to recover for 14 d before participating in further behavioral training. We allowed 4–6 wk for viral expression before commencing optogenetic experiments.

## Neural recordings

Neural recordings were performed using resin-coated tungsten electrodes (impedance ~ 300 kΩ, FHC, Bowdoin, ME). Electrodes were attached to a head stage (Cereplex, Blackrock Microsystems, Salt Lake City, UT), then digitized using a Cerebus (Blackrock Microsystems) and stored on a PC for offline analysis. For single and multiunit analysis, the neural signals were band-pass filtered between 300 and 8000 Hz, and the LFP data was low-passed at 300 Hz, spikes were detected using Offline Sorter (Plexon, Dallas, TX), and all data were analyzed using custom routines in MATLAB (MathWorks, Natick, MA). For optogenetic activation of the LGN, we employed an optrode, that is, a 100-µm-diameter optic fiber coupled to the same tungsten electrodes described above for simultaneous monitoring of neural activity close to light stimulation site. The optic fiber was connected to a 473 nm blue laser (Changchun New Industries Optoelectronics, China). Laser intensity was set to 3.5 mW using an optical power meter (PM 100D, Thorlabs Newton, NJ). Stimulation was delivered using a duty cycle of 50% controlled by a pulse generator (Rigol, Beaverton, OR). Visual stimulation was delivered on a video monitor (VPixx, Canada), placed 28.5 cm from the animal and covering the visual field location corresponding to the receptive field of the neurons being recorded, as described previously (*Veit et al., 2014*). For the functional validation experiments (*Figure 2*), we analyzed activity of 120 single neurons in the LGN obtained in n = 4 tree shrews. For the functional coupling and decoding analyses in LGN and V1 (*Figures 6–8*), we analyzed simultaneous LFP and MUA recordings obtained at 111 sites in LGN and V1 in n = 4 tree shrews. In both cases, CamKIIα-ChR2 injections were made in the LGN 4–6 wk prior to the experiment. We generally employed 40 Hz light stimulation frequency, and a subset of recordings were made using a range of frequencies (30, 40, 60, and 80 Hz). Details regarding the number of recordings obtained in each animal are provided in *Table 3*. For V1 LFP spectral signal detection analysis, we extract for each single trial the Fourier spectral amplitude (SA) at the stimulation frequency ($f_{stim}$ Hz) and control values at $f_{stim}$-3 Hz during baseline and opto-stimulation conditions. We then computed hit rate as the fraction of trials for which $SA_{opto}(f_{stim}) > SA_{opto}(f_{stim}-3)$, that is, hits/(hits+misses) during opto-stimulation, and the false alarm rate as the fraction of trials for the spectral peak at stimulation frequency during baseline exceeded the corresponding value during optical stimulation, that is, $SA_{baseline}(f_{stim}) > SA_{opto}(f_{stim})$.

## Spike-triggered average and spike field coherence

In order to calculate the STA, we averaged 200 ms epochs of V1 LFPs centered on the LGN spikes. For the SFC, we took the FFT of the STA, and then divided it by the average spectral power calculated from the FFTs of all the epochs used in calculated the STA, $\bar{P}$. In this way, the SFC is normalized to the existing power in the cortical LFP. $SFC(f) = \left[\frac{STAfft(f)}{\bar{P}(f)}\right]$, where $f$ is an individual frequency band.

**Table 3.** Summary of electrophysiological recordings.

| Tree shrew | Total MUA and LFP recordings LGN 40 Hz | Significant MUA entrainment at 30/40/60/80 Hz LGN | Significant 40 Hz 'ON' and 'OFF' responses in LGN | Significant 40 Hz LFP entrainment in V1 | Significant LFP entrainment at 30/40/60/80 Hz in V1 |
|---|---|---|---|---|---|
| Figure panel | | *Figure 2* | *Figure 6A/B* | *Figure 7A* | *Figure 8B* |
| 207 | 17 | Not studied | 8/5 | 1 | Not studied |
| 1503 | 47 | Not studied | 4/2 | 3 | Not studied |
| 1903 | 30 | 2/1 / 1/0 | 1/2 | 12 | 1/0/2/3 |
| 111 | 17 | 7/9 / 9/10 | 9/7 | 13 | 2/9/9/10 |
| Total | 111 | 9/10 / 10/10 | 22/16 | 29 | 3/9/11/13 |

LFP, local field potential; LGN, lateral geniculate nucleus; MUA, multi-unit activity.

## Vector strength

Vector strength was calculated as $R = \frac{1}{N} \left| \sum_{j=1}^{N} 1^{e^{i\theta_j}} \right|$ , where $R$ is the vector strength, $N$ is the number of trials, and $\theta_j$ is the phase angle of the spike in relation to the flicker stimulus being delivered, 30, 40, 60, or 80 Hz (*Goldberg and Brown, 1969*). Significance was then calculated using the Rayleigh test statistic.

## Histology

The tree shrew (*T. belangeri*) was deeply anesthetized with 100 mg/kg pentobarbital, then perfused transcardially with 400 ml 0.1 M phosphate-buffered saline (PBS) (pH 7.4) followed by 400 ml cold 4% paraformaldehyde in 0.1 M PBS. Whole brains were placed in the same fixative overnight at 4°C, then rinsed three times for 20 min in cold 0.1 M PBS and cryoprotected by immersion in a sucrose gradient (15 and 30% w/v sucrose) until it sank. Brains were blocked and fast frozen in dry-ice chilled isopentane and stored at least overnight at –20°C before proceeding with the cryosectioning. At that point, brains were sectioned into 40-µm-thick coronal sections using a sliding microtome (MICROM HM 440E, Microm International GmbH, Walldorf, Germany). Every third section was preserved in a storage solution (30% ethylene glycol and 30% glycerol in 0.1 M phosphate buffer) at –20°C. To visualize the virus-mediated expression of CaMKIIα-Chr2-mCherry by fluorescence, sections were mounted on Superfrost Plus Adhesion slides (Fisher Scientific AG, Reinach, CH), and cover-slipped using an aqueous Vectashield antifade mounting media with Dapi (Vector Laboratories, Inc, Burlingare, CA, H-1200). Virus expression was imaged using a NanoZoomer 2.0-HT slide scanner (Hamamatsu Photonics), with a 2 × 20 0.75 NA air objective and at a resolution of 0.23 µm/pixel. Visualization and analysis were executed with the NDP.view 2 freeware (Hamamatsu Photonics).

## Immunohistochemistry

We performed immunohistochemistry for the specific neuronal markers parvalbumin (PV) and CaMKIIα in the LGN. Cryopreserved free-floating sections were previously washed by rinsing in 0.1 M PBS five times for 20 min. Then sections were subjected to heat-induced antigen retrieval (HIER) in a Tris-EDTA-based solution at pH 9.0. Sections were then permeabilized for 110 min with 0.1 M PBS +0.3% Triton X-100, followed by incubation in blocking solution for 2 hr at room temperature, containing 0.1 M PBS-0.05% Triton X-10, 0.3 M glycine, and 15% normal donkey serum (NDS) (Abcam, AB 7475). Next, sections were incubated with goat anti-CamKIIα polyclonal antibody (3.3 µg/ml; Invitrogen, Thermo Fisher Scientific, Inc, Regensburg, DE, PAS-19128) and mouse anti-PV (1:5000; Swant, Switzerland; 235Pur) in 0.05% Tween 20, 7% NDS, and 0.02% NaN₃ in 0.1 M PBS for 48 hr at 4°C. Sections were washed in 0.1 M PBS + 0.05% Tween 20 + 1% NDS five times, 20 min each, and then incubated with the different fluorescent secondary antibodies diluted in 0.05% Tween-20, 1% NDS, and 0.1 M PBS for 16 hr at 4°C. The secondary antibodies were Alexa Fluor 488 donkey anti-mouse IgG (H+L) (1:500; Jackson ImmunoResearch, Europe Ltd, UK; Cat# 715-545-151), Alexa Fluor 647 AffiniPure donkey anti-goat IgG (H+L) (1:500; Jackson ImmunoResearch, Cat# 705-605-147). Nuclei were counterstained with DAPI at 0.5 µg/ml for 5 min. Sections were mounted on Superfrost Plus Adhesion slides (Fisher Scientific AG) and cover-slipped with 90% glycerol + 0.5% N-propyl gallate in 20 mM Tris at pH 8.4. Images were collected using a ×20 NA 0.75 mm multi-immersion objective on a Leica STELLARIS SP8 FALCON Laser scanning confocal microscope (Leica Microsystems AG, Switzerland) and analyzed using LAS X software (version 3.3.0, Leica Microsystems). Image processing was executed with the ImageJ/Fiji software (NIH, Bethesda, MD).

## Additional information

### Funding

| Funder | Grant reference number | Author |
| --- | --- | --- |
| Swiss National Science Foundation | 182504 | Gregor Rainer |
| University of Fribourg | | Gregor Rainer |

| Funder | Grant reference number | Author |
|--------|------------------------|--------|

The funders had no role in study design, data collection and interpretation, or the decision to submit the work for publication.

## Author contributions

Jing Wang, Formal analysis, Investigation, Writing – original draft; Hamid Azimi, Data curation, Software, Formal analysis, Investigation, Writing – original draft; Yilei Zhao, Melanie Kaeser, Investigation; Pilar Vaca Sánchez, Formal analysis, Investigation, Methodology; Abraham Vazquez-Guardado, Resources, Methodology; John A Rogers, Resources, Supervision, Funding acquisition, Methodology; Michael Harvey, Conceptualization, Software, Formal analysis, Writing – original draft, Writing - review and editing; Gregor Rainer, Conceptualization, Data curation, Software, Formal analysis, Supervision, Funding acquisition, Methodology, Writing – original draft, Project administration, Writing - review and editing

## Author ORCIDs

Yilei Zhao (iD) http://orcid.org/0000-0002-5146-8493
Abraham Vazquez-Guardado (iD) https://orcid.org/0000-0002-0648-5921
Michael Harvey (iD) http://orcid.org/0000-0002-0477-6100
Gregor Rainer (iD) http://orcid.org/0000-0002-5805-2220

## Ethics

All procedures for animal experiments were approved by the local ethical committee on animal experimentation, canton of Fribourg. License number: 33056.

## Decision letter and Author response

Decision letter https://doi.org/10.7554/eLife.90431.sa1
Author response https://doi.org/10.7554/eLife.90431.sa2

## Additional files

### Supplementary files

• MDAR checklist

### Data availability

All data used in the production of the figures in this manuscript is freely available at https://doi.org/10.5061/dryad.2z34tmpqk.

The following dataset was generated:

| Author(s) | Year | Dataset title | Dataset URL | Database and Identifier |
|-----------|------|---------------|-------------|-------------------------|
| Rainer G | 2023 | Optogenetic activation of visual thalamus generates artificial visual percepts | https://datadryad.org/stash/dataset/doi:10.5061/dryad.2z34tmpqk | Dryad Digital Repository, 10.5061/dryad.2z34tmpqk |

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
