## [Editor Report]

This important study shows that tree shrews can detect optogenetic stimulation of the lateral geniculate nucleus (LGN) after training detection of visual stimuli. The solid evidence links optogenetic stimulation of the LGN to behavioral detection and neurophysiological responses. This article is potentially of interest to neuroscientists and clinicians working on the visual system and visual prostheses.

---

## [Decision Letter]

**Decision letter after peer review:**

[Editors’ note: the authors submitted for reconsideration following the decision after peer review. What follows is the decision letter after the first round of review.]

Thank you for submitting the paper "Optogenetic activation of visual thalamus generates artificial visual percepts" for consideration by *eLife*. Your article has been reviewed by 3 peer reviewers, including Kristine Krug as the Reviewing Editor and Reviewer #1, and the evaluation has been overseen by a Senior Editor.

Comments to the Authors:

We are sorry to say that, after consultation with the reviewers, we have decided that this work will not be considered further for publication by *eLife*.

While the reviewers found that the demonstration in the first part of the manuscript that optogenetic stimulation in the LGN of tree shrews can be detected is interesting and novel, they agreed that both parts of the study suffer from significant limitations. The critical link between the two parts also seems to be missing – with the investigation of the discrimination of the visual flicker frequencies being valuable but not really connected to the optogenetic experiment.

All three reviewers felt that the manuscript would be decidedly stronger if their concerns about the main optogenetic experiment (part 1) were addressed either as a self-contained study or with all concerns addressed plus the clearly missing link of an optogenetic frequency discrimination study included in part 2. But we appreciate that in both cases this would be a different manuscript and is beyond a major revision.

*Reviewer #1 (Recommendations for the authors):*

(1) The two parts of the manuscript – detection/optogenetics and flicker discrimination/recordings – seem not well connected and rather seem to belong to two different studies. Different optogenetic stimulation frequencies were clearly employed in the behavioural assessment of optogenetics (Figure 4B) and they are also encoded by LGN neurons (Figure 2). But the link is not addressed here, so the manuscript appears to contain two separate studies.

(2) There seems to be some experimental detail missing or unclear in the results (and methods), importantly the number of animals used for each part, for individual animals an ID (so different parts can be connected more easily), and in some parts summary statistics across all experiments carried.

For example:

p5/6: how many animals were injected, on how many of these were immunohistochemistry done, and how many cells were counted across how many animals and sections? Please add from how many animals, you recorded how many neurons.

Figure 1: provide section plane and give animal ID – which of your behavioural study animals is this from, please?

Figure 2: give animal ID, which of your behavioural study animals is this from, please?

Line 140-142: how many neurons did your record in total and how did you define a burst

Figure 3: in D, there seem to be 5 animals, in E-G only 4. Can you add the green animals in or if not, why not. Please provide animal IDs for the 5 colour-coded animals in Figures 3+4, so they can be linked to specific animal data in the previous section.

Line 228 and methods: Please explain somewhere what a VI-30 schedule is.

Methods: please add how many animals in total were used and what were the histology protocols.

A table in the methods would be helpful: for each animal, it should show what training and experiments, it took part in and in what order.

(3) Since single combinations of strength and frequency of stimulation were used in each session and session order randomised, it would be helpful to show (i) for each animal in the methods, the sequence of experiments, (ii) for each animal in parallel d' over sessions in order of experiments. To exclude effects of training and/or "wearing out" of stimulation sites over time.

4) Flicker results – line 268 onwards.

Please specify how many animals were involved and that these were anaesthetised experiments.

Please distinguish between encoding in firing rate or temporal pattern of spiking.

I might be missing something but in 7B, there is a clear peak for 40 Hz visual stimulation in the evoked pattern, however, in the third panel on the right, no peaks are shown for 40 Hz. Please explain.

Lines 299-302: How many cells from how many animals were recorded in total, and what fraction does that constitute for each animal?

(5) Can you provide a direct comparison (in a figure perhaps) of the best d' during optogenetics and the best d' in the visual detection task, animal-by-animal?

(6) Lines 437-438: Please provide details on which regulations and which authorities.

*Reviewer #2 (Recommendations for the authors):*

I believe that the study can be significantly strengthened by reporting whether animals that were trained in the visual flicker frequency discrimination task are able to transfer to optogenetic frequency discrimination. How is this information encoded among LGN and the downstream V1 neurons (similar to Figures7 and 8)? What are the similarities and differences between visual flicker frequency representation and optogenetic frequency representation?

*Reviewer #3 (Recommendations for the authors):*

Specific comments:

1. What is the rationale for testing for co-expression of ChR2/mCherry and PV? What fraction of the thalamocortical neurons in tree shrew LGN express PV and what fraction of PV cells also express CaMKIIs? Was histology done in more than one animal? Are there additional regions beyond the example in Figure 1 where immunostaining had been performed? As it stands, this portion of the paper is anecdotal and its significance is unclear.

2. How many animals have been studied with electrophysiology and optogenetics and how many single units have been recorded? A summary analysis beyond the three example cells in Figure 2 would be helpful. It would be nice to include PSTHs in Figure 2A in addition to the power spectrum. Can the authors demonstrate optogenetic stimulation and characterization of ON/OFF visual responses in the same cells? How were bursts defined?

3. The way that the detection task is analyzed is confusing, and while this approach may be justifiable, it needs to be explained more clearly. The visual/optogenetic target appears on every trial but some hits and some false alarms are assigned as correct rejects depending on when the target appeared (Figure 3B). It is unclear whether these somewhat arbitrary categories also impacted the reward delivery to the animals. The day-to-day variability in behavioral performance appears to be quite high so analyses such as the assessment of the effect of eccentricity on detectability require multiple sessions per eccentricity per animal. This is true throughout this manuscript. A single behavioral session per animal may not be sufficient for making meaningful inferences. Quantitative analysis of the distribution of response times would be beneficial. The results could then be compared with the response times during the optogenetic stimulation.

4. Demonstration of the detection of optogenetic stimulation in the LGN in Figure 4 is the central finding in this paper but the results are quite anecdotal. Is each data point in 4B a single session? If so, these results may not be interpretable and additional sessions/animals are needed in order to make inferences about the effect of the optogenetic stimulation parameters. It is unclear why there are 5 animals in Figure 4 and only 4 in Figure 3. Were all animals tested in optogenetic detection previously trained in visual detection? Where were the off-LGN injections done in the control animals? Demonstrating the effect in more than three animals could strengthen the paper. As in visual detection, it would be helpful to see quantitative analysis of the distribution of response times with comparison across animals and stimulation parameters.

5. It would be interesting to analyze performance as a function of time during the first optogenetic stimulation session per animal; does the probability of reporting the stimulation increase during the first session?

6. The description of the flicker frequency discrimination task is unclear. Many readers will not know what operant schedules and VI30 are. What is the relation between nose pokes, lever presses, and rewards? Are the blue tick marks in Figure 5A-B rewards or lever presses? It is not clear how the four behavioral categories were defined.

7. It would be nice to show data from all 6 animals in Figure 6A since summary data from all 6 is shown in panels B-D. How can mean response latency in S+ be between 25 and 45 sec (Figure 6C) but response frequency between 0.1 and 0.2 pokes/sec (Figure 6D)?

8. Were the data for the analysis of flicker frequency discrimination sensitivity by LGN and V1 neurons collected from anesthetized animals? Were these acute experiments? How many animals have been studied? How many of the neurons were ON vs. OFF dominated? The analysis of the frequency decoder based on the peak of the FFT is suboptimal. A decoder can rely on power at all frequencies to discriminate stimulus flicker frequency.

9. I may have missed these details but I did not see a description of the immunostaining methods. Were the μ‐LEDs implanted immediately after the viral injection? how close were they to the viral injection site? How were they inserted? For how long did they have an effect? Was the tissue analyzed for possible damage after the experiments were completed?

10. Adding section subheadings could help readability. In general, the paper could benefit from more careful editing.

[Editors’ note: further revisions were suggested prior to acceptance, as described below.]

Thank you for submitting your article "Optogenetic activation of visual thalamus generates artificial visual percepts" for further consideration by *eLife*. Your revised article has been evaluated by Kristine Krug as Reviewing Editor and Andrew King as the Senior Editor.

The manuscript has been improved but there are some remaining issues that need to be addressed, as outlined below.

Essential revisions:

(1) Summary data for the example in Figure 2 to be able to generalise the results.

2) A description of the histological methods used.

3) Alignment of RFs in LGN and V1. Could this explain the low evoked spiking activity in V1?

(4) The reviewers requested a number of essential experimental details and some changes in the presentation.

*Reviewer #1 (Recommendations for the authors):*

(1) Please expand Table 1 to include the other four animals for neurophysiology and histology. Include in particular the histological data (Figure 1 and supplement) and the neurophysiology results, e.g. how many neurons and LFP recordings with significant responses were recorded from each animal.

The former is informative to link the different expression patterns across layers to the shown results. I appreciate this is incomplete, but would still be useful.

The latter is essential to understand whether the significant responses come just from one or more animals.

(2) Given that the response letter stated that the histological protocols had been included in the revised methods, it was disappointing to find it still missing. For the histological data to be appropriately documented for a journal paper, details and timing of the perfusion need to be included as well as the sectioning and the staining protocols.

(3) I assume Figure 2 is the example from one animal. Please identify the animal and provide additional data to what extent this result was validated across how many animals and how many cells from each animal.

(4) The low optogenetically evoked MUA responses especially in V1 are surprising. Could the authors show the alignment of the RFs mapped for the optogenetic injection site and for the subsequent cortical recording sites for the animals and link them to the results in each animal, please?

5) Figure 2B figure legend: add that this is visual stimulation, please.

6) From which animal is the performance shown in Figures 3A and 4B? Please add to the figure or figure legend.

(7) sFigure 6: For animal 1409, it would be helpful to explain to the reader how a performance of that is generally below 50% correct leads to a d' >1 on the first day of optogenetic stimulation. Unless I am misunderstanding the two graphs?

*Reviewer #3 (Recommendations for the authors):*

The paper is much improved, and the focus on the behavioral effects of LGN optogenetic stimulation strengthens the paper. While the basic finding seems to be robust, some details are not described adequately, several aspects of the presentation can be improved, and the manuscript could benefit from more careful editing.

Specific comments

– Figure 2 is nice but this is only one example. It is important to include a summary figure to follow up on Figure 2 that shows quantitatively how key findings from Figure 2 generalize to all recorded cells. For example, do most cells phase lock up to 60 Hz? What is the overall firing rate as function of stimulation frequency?

– Please change y–axis of PSTHs in Figures2 and 6 to firing rate.

– Figure 4B – should be mW and not W in the key.

– In Figure 4 it would be helpful to add an icon or text in B to indicate which animals are controls.

– In Figure 5A, it would be helpful to add a panel that shows d' per session as a function of session # for each animal. The session–to–session variability is important to assess in order to interpret the effects of power and frequency in Figure S6 since each combination was only used once per animal.

– Figure 6 is missing panel labels.

– 5 units in V1 showing significant change in firing rate during optostim at p<0.05 out of 111 is expected by chance and the modulation according to Figure 6A is very modest; a more appropriate description of these results is that the authors failed to see V1 effects in the single/multi–unit level.

– In Figure 8B, each set of connected symbols is a different site. If so, can you comment on why the effect is so bimodal? Are filled symbols significant optostimulation effect?

– Why compute STA of LFP in V1 based on LGN spikes rather than on based LGN optogenetic stimulation? Is there a significant difference between these two?

– Was there any indication to the animal of a miss other than withdrawal of the reward? Was the intertrial interval fixed? Was the ITI affected by misses?

– The text discussing "on" and "off" cells is a little confusing. It would be better to say explicitly that "on" and "off" visually responsive cells are segregated in the tree shrew LGN layers and refer to the supplementary figure.

– The terminology may be confusing to the readers. Instead of "light stimulation", perhaps say "visual stimulation" to better contrast with optogenetic stimulation?

– It would be helpful to indicate in the figures which panels show results from optogenetic stimulation and which from visual stimulation.

– "as has been reported also for humans (Erickson–Davis and Korzybska, 2021)" may be interpreted by the reader as implying that optogenetic stimulation has been done in humans but this is an electrical stimulation study.

– "it has been shown that for macaques, extensive training on optogenetic stimulus detection in V1 can be detrimental for detection of visual stimuli (Ni and Maunsell, 2010)" – this paper used microstimulation and not optogenetic stimulation.

---

## [Author Response]

[Editors’ note: the authors resubmitted a revised version of the paper for consideration. What follows is the authors’ response to the first round of review.]

Comments to the Authors:Reviewer #1 (Recommendations for the authors):(1) The two parts of the manuscript – detection/optogenetics and flicker discrimination/recordings – seem not well connected and rather seem to belong to two different studies. Different optogenetic stimulation frequencies were clearly employed in the behavioural assessment of optogenetics (Figure 4B) and they are also encoded by LGN neurons (Figure 2). But the link is not addressed here, so the manuscript appears to contain two separate studies.

We thank the reviewer for this comment, which was also brought up by an additional reviewer. We have now completely revised the manuscript, such that the flicker discrimination is no longer included and we focus entirely on the optogenetic detection. We have added substantial additional behavioral data, histological analyses and neural circuit studies related to information processing in LGN and V1; all of which support our results in the first part of the original manuscript. We feel this has made the manuscript far more cohesive.

(2) There seems to be some experimental detail missing or unclear in the results (and methods), importantly the number of animals used for each part, for individual animals an ID (so different parts can be connected more easily), and in some parts summary statistics across all experiments carried.

We have now provided 2 additional tables detailing the animals, and their IDs, used in each part of the experiments. Tables 1 and 2 have been added to the methods, and we also provide additional statistical validation.

For example:p5/6: how many animals were injected, on how many of these were immunohistochemistry done, and how many cells were counted across how many animals and sections? Please add from how many animals, you recorded how many neurons.

We now provide animal numbers where appropriate in each section. We have also completely restructured the immunohistochemistry. We now demonstrate virus expression in the LGN in 2 additional animals (supp.Figure 2) for a total of 3 animals. We have added completely novel histology for PV and CamKIIα (Figure 1A: 1 animal, supp.Figure 1: 2 animals), demonstrating robust CaMKIIexpression across LGN layers. The PV staining is added as it is useful for layer delineation. For the functional validation neurophysiology (Figure 2), we analyzed data from 120 single neurons obtained in the LGN, focusing on single neuron activity as this is necessary for burst analysis. For the functional interaction neurophysiology (Figure 6‐8), we recorded from 111 sites in LGN and V1, focusing on multi‐unit‐activity (MUA) here to optimize the probability of detecting functional relationships between remote regions.

Figure 1: provide section plane and give animal ID – which of your behavioural study animals is this from, please?

We have provided the plane of the section in the new figure 1 as well as the related supplementary figure 2. As mentioned in the legend of supplementary Figure 2, the top row viral expression is from animal 1806, which participated in the behavioral study.

Figure 2: give animal ID, which of your behavioural study animals is this from, please?

Figure 2 has now been combined with Figure 1, and data is available for tree shrew 1806 in suppl. Figure 2.

Line 140-142: how many neurons did your record in total and how did you define a burst

We have added this information to the methods section: “We recorded from 120 single units in the LGN, and defined a burst as two or more action potentials with inter‐spike intervals <4ms, and preceded by at least 50 ms without spiking activity.”

Figure 3: in D, there seem to be 5 animals, in E-G only 4. Can you add the green animals in or if not, why not. Please provide animal IDs for the 5 colour-coded animals in Figures 3+4, so they can be linked to specific animal data in the previous section.

We have updated figures 3 and 4, and the new figure 5, to show the animal IDs. The green animal, animal 1409, did not participate in the spatial generalization test, thus it is not included in that set of results. We felt that four animals are sufficient for demonstration of visual spatial generalization, and have also added a statistical validation of this finding.

Line 228 and methods: Please explain somewhere what a VI-30 schedule is.

VI‐30 task is no longer present in the manuscript.

Methods: please add how many animals in total were used and what were the histology protocols.

We have now included the total number of animals, and the protocols for the immunohistochemistry.

A table in the methods would be helpful: for each animal, it should show what training and experiments, it took part in and in what order.

A table (table 1) has been added to the methods.

(3) Since single combinations of strength and frequency of stimulation were used in each session and session order randomised, it would be helpful to show (i) for each animal in the methods, the sequence of experiments, (ii) for each animal in parallel d' over sessions in order of experiments. To exclude effects of training and/or "wearing out" of stimulation sites over time.

We thank the reviewer for this insightful comment. For these experiments, we used the same serial order of conditions for all participating animals, as now shown in table 2 (methods section). We have also added supplementary Figure 6, which shows that “wearing out” does not account for the observed effects.

(4) Flicker results – line 268 onwards.Please specify how many animals were involved and that these were anaesthetised experiments.Please distinguish between encoding in firing rate or temporal pattern of spiking.I might be missing something but in 7B, there is a clear peak for 40 Hz visual stimulation in the evoked pattern, however, in the third panel on the right, no peaks are shown for 40 Hz. Please explain.Lines 299-302: How many cells from how many animals were recorded in total, and what fraction does that constitute for each animal?

This data is no longer included in the manuscript.

(5) Can you provide a direct comparison (in a figure perhaps) of the best d' during optogenetics and the best d' in the visual detection task, animal-by-animal?

We appreciate this suggestion and have now included this information in supp Figure 4.

(6) Lines 437-438: Please provide details on which regulations and which authorities.

This information now appears at the beginning of the methods section.

“The local ethical committee on animal experimentation (canton of Fribourg) approved all experimental procedures.”

Reviewer #2 (Recommendations for the authors):I believe that the study can be significantly strengthened by reporting whether animals that were trained in the visual flicker frequency discrimination task are able to transfer to optogenetic frequency discrimination. How is this information encoded among LGN and the downstream V1 neurons (similar to Figures7 and 8)? What are the similarities and differences between visual flicker frequency representation and optogenetic frequency representation?

We have now removed everything related to visual flicker frequency discrimination from the manuscript, as was suggested also by the other reviewers. As suggested by this reviewer, we have included analyses related to the transfer between visual and optogenetic paradigms, focusing on the detection task. The remaining comments of the reviewer are highly appreciated; we are indeed planning to follow up the visual flicker experiment with an optogenetic flicker detection variant, that will allow us to examine the similarities and differences in terms of temporally structured stimulation. For the revised manuscript, we focus on the initial proof of principle demonstration using a detection paradigm. We have however added substantial neurophysiology results, that make a strong neural circuit case for optogenetically induced functional activations in LGN and (more importantly) V1.

Reviewer #3 (Recommendations for the authors):Specific comments:1. What is the rationale for testing for co-expression of ChR2/mCherry and PV? What fraction of the thalamocortical neurons in tree shrew LGN express PV and what fraction of PV cells also express CaMKIIs? Was histology done in more than one animal? Are there additional regions beyond the example in Figure 1 where immunostaining had been performed? As it stands, this portion of the paper is anecdotal and its significance is unclear.

We agree with the reviewer. We have now performed dual immunostaining for CaMKII and parvalbumin in the LGN of 3 tree shrews, see the new figure 1 and supplementary figure 1. While in the monkey CaMKIIexpression is restricted to the interlaminar regions of the LGN, we show here that in the tree shrew, it expressed throughout the entire extent of LGN including the laminae proper. With this finding, the cell counting and relation to PV projection system becomes less important, so we have in fact removed cell counting related to co‐expression of PV and CaMKII We have however done PV staining also for the new animals, as this permits a particularly clear delineation of the LGN layers. The co‐expression analysis can be done, but we feel that it is not crucial to the scientific message of this particular paper and will be important when more specific LGN populations are targeted for optogenetics in future studies. In addition, we also show thalamocortical axons in V1 in two additional animals (suppl. Figure 2), corroborating therefore both points (robust CaMKII expression in LGN and thalamocortical fibers) in a total of 3 animals each.

2. How many animals have been studied with electrophysiology and optogenetics and how many single units have been recorded? A summary analysis beyond the three example cells in Figure 2 would be helpful. It would be nice to include PSTHs in Figure 2A in addition to the power spectrum. Can the authors demonstrate optogenetic stimulation and characterization of ON/OFF visual responses in the same cells? How were bursts defined?

We have now included the number of animals in the *Subjects* subsection of the methods. We have stated the number of units and the definition of a burst. “*We recorded from 120 single units in the LGN, and defined a burst as two or more action potentials with inter‐spike intervals <4ms, and preceded by at least 50 ms without spiking activity.”* The new figures 6,7&8 now provide summaries of the recorded neurons. We have included PSTHs in figure 2A. We do not however have sufficient data at this point for a thorough characterization of ON‐ and OFF‐response optogenetics, and have included this aspect as a validation of previously published findings.

3. The way that the detection task is analyzed is confusing, and while this approach may be justifiable, it needs to be explained more clearly. The visual/optogenetic target appears on every trial but some hits and some false alarms are assigned as correct rejects depending on when the target appeared (Figure 3B). It is unclear whether these somewhat arbitrary categories also impacted the reward delivery to the animals. The day-to-day variability in behavioral performance appears to be quite high so analyses such as the assessment of the effect of eccentricity on detectability require multiple sessions per eccentricity per animal. This is true throughout this manuscript. A single behavioral session per animal may not be sufficient for making meaningful inferences. Quantitative analysis of the distribution of response times would be beneficial. The results could then be compared with the response times during the optogenetic stimulation.

The reviewer makes important points. We have added some more information on the detection task analysis using signal detection framework (methods section). The advantage of this approach is that a single value (d’ sensitivity) can be computed for each behavioral session, which takes into account errors of both categories (false alarm and misses). Our approach has the key advantage that we do not require any signal absent trials, which would be problematic for animal motivation particularly in tree shrews. Rewards are delivered only after detected targets, and the analysis framework had no impact on the reward schedule. The reviewer is correct that such an effect would be problematic, but our analysis is designed to expressly avoid this.

Relating to variability, it is true that there is considerable day‐to‐day variability and statistical reproducibility is key to solid inference. We have therefore conducted 30 sessions of optogenetic detection experiments in animals 959, 1111, 1409 to corroborate our findings and added an additional animal 1806 (Figure 5), and these now we feel unequivocally show that for the optimal stimulation parameters (determined in pilot experiments see Figure 4), tree shrews can detect the optogenetic stimulation. We have also added statistical analysis on the visual detection task, showing that detection performance (d’) for all animals was significantly different from zero.

For the eccentricity effects, we have added a statistical test comparing d’ across the studied eccentricities against zero; a test that is significant for all four animals studied. We have added this information to the manuscript. Please note that d’ sensitivity is itself a summary statistic based on 200 trials of various types, and in this setting, we consider estimates on single session to be meaningful.

4. Demonstration of the detection of optogenetic stimulation in the LGN in Figure 4 is the central finding in this paper but the results are quite anecdotal. Is each data point in 4B a single session? If so, these results may not be interpretable and additional sessions/animals are needed in order to make inferences about the effect of the optogenetic stimulation parameters. It is unclear why there are 5 animals in Figure 4 and only 4 in Figure 3. Were all animals tested in optogenetic detection previously trained in visual detection? Where were the off-LGN injections done in the control animals? Demonstrating the effect in more than three animals could strengthen the paper. As in visual detection, it would be helpful to see quantitative analysis of the distribution of response times with comparison across animals and stimulation parameters.

We thank the reviewer for this comment, and now report data from multiple session for each of the animals using the optimal frequency and (high) amplitude (Figure 5). We feel that these data now unequivocally demonstrate that optogenetic stimulation is detected by the tree shrews. The experiments with multiple frequency and amplitudes have the character of pilot data used for selecting stimulation parameters for the study on repeatability of the effects. This type of pilot phase is typically done in this type of experiment for parameter selection; but often not reported in detail. Due to considerations including animal motivation, implant lifetime and efficacy, 3R considerations we limited the total number of behavioral sessions during which we were able to study each animal. We also now report the results of the amplitude/frequency combination experiment in each animal in the serial order that they were done, providing a full characterization of the experimental findings.

5. It would be interesting to analyze performance as a function of time during the first optogenetic stimulation session per animal; does the probability of reporting the stimulation increase during the first session?

In Figure 3, 6 animals learned the initial task (panels C,D); as we have added an additional animal since the initial submission. For (visual) spatial generalization was tested in only four of these animals. We felt that four animals are sufficient to demonstrate spatial generalization, and now provide a statistical test demonstrating significance of the generalization performance.

The control injections were performed in 2 animals in the regions of the globus pallidus and zone incerta. The coordinates were: anterior posterior from the interaural line = 6.11; medial lateral = 4; dorsal ventral = 8.5. Of the 6 animals tested on visual detection, 4 were implanted in the LGN and 2 outside the LGN.

We have done the response time analysis for the visual detection task over the course of training as requested by the reviewer (supplementary Figure 7); the findings suggest that improved d’ sensitivity is accompanied by trends towards speeded (animal 959) or slowed (animal 1111) reaction times, with some apparent variability between individual tree shrews.

6. The description of the flicker frequency discrimination task is unclear. Many readers will not know what operant schedules and VI30 are. What is the relation between nose pokes, lever presses, and rewards? Are the blue tick marks in Figure 5A-B rewards or lever presses? It is not clear how the four behavioral categories were defined.

We have added this information in supplementary figure 6. The data shows that animal 1111 already correctly reported at high performance on the initial trials (data in figure is smoothed by convolution across 10 trials), and tree shrews tended to show an increasing performance within the session consistent with a within‐session learning effect.

7. It would be nice to show data from all 6 animals in Figure 6A since summary data from all 6 is shown in panels B-D. How can mean response latency in S+ be between 25 and 45 sec (Figure 6C) but response frequency between 0.1 and 0.2 pokes/sec (Figure 6D)?8. Were the data for the analysis of flicker frequency discrimination sensitivity by LGN and V1 neurons collected from anesthetized animals? Were these acute experiments? How many animals have been studied? How many of the neurons were ON vs. OFF dominated? The analysis of the frequency decoder based on the peak of the FFT is suboptimal. A decoder can rely on power at all frequencies to discriminate stimulus flicker frequency.

These comments refer to the flicker experiment, which has been removed from the manuscript. However, we take these into account in preparation of a manuscript on these data and thank the reviewer for these comments.

9. I may have missed these details but I did not see a description of the immunostaining methods. Were the μ‐LEDs implanted immediately after the viral injection? how close were they to the viral injection site? How were they inserted? For how long did they have an effect? Was the tissue analyzed for possible damage after the experiments were completed?

We thank the reviewer for pointing this out. We have now included a full description of the immunohistochemistry in the methods section. Yes, the ‐LEDs were implanted immediately following viral injection, with their coordinates matching those of the injection site. This information is detailed in the methods under *surgical procedures.* LEDs were all still functional when the animals were killed. However, we have not done a longitudinal study of their efficacy over longer periods than what was useful for the study. The Nissl‐stained tissue was analyzed under the microscope, and we found no visually detectable damage.

10. Adding section subheadings could help readability. In general, the paper could benefit from more careful editing.

We thank the reviewer for this suggestion, and have now included subheadings. We have made a number of editorial changes, including removing everything pertaining to the flicker discrimination study, as suggested also by another reviewer. We feel that this has greatly improved the flow and lucidity of the manuscript.

[Editors’ note: what follows is the authors’ response to the second round of review.]

The manuscript has been improved but there are some remaining issues that need to be addressed, as outlined below.Essential revisions:(1) Summary data for the example in Figure 2 to be able to generalise the results.

We have performed additional analyses related to figure 2, and now also report group data in addition to the single example.

(2) A description of the histological methods used.

We have added this to the manuscript.

(3) Alignment of RFs in LGN and V1. Could this explain the low evoked spiking activity in V1?

Yes; we already mentioned this possibility in connection with the LFP analyses (Fig8) but have added a sentence also in the Results section related to Figure 6.

(4) The reviewers requested a number of essential experimental details and some changes in the presentation.

We have provided the additional details and presentation changes requested by the reviewers.

Reviewer #1 (Recommendations for the authors):(1) Please expand Table 1 to include the other four animals for neurophysiology and histology. Include in particular the histological data (Figure 1 and supplement) and the neurophysiology results, e.g. how many neurons and LFP recordings with significant responses were recorded from each animal.The former is informative to link the different expression patterns across layers to the shown results. I appreciate this is incomplete, but would still be useful.The latter is essential to understand whether the significant responses come just from one or more animals.

We have added table 3 in the methods section under “neural recordings”, p.24, l. 616 which details how many recordings were made in each animal and also for major analyses in which animals significant effects were found.

(2) Given that the response letter stated that the histological protocols had been included in the revised methods, it was disappointing to find it still missing. For the histological data to be appropriately documented for a journal paper, details and timing of the perfusion need to be included as well as the sectioning and the staining protocols.

We thank the reviewer for pointing out this omission and have added this information to the methods section, pp. 25-26, ll. 629-669

(3) I assume Figure 2 is the example from one animal. Please identify the animal and provide additional data to what extent this result was validated across how many animals and how many cells from each animal.

In the original submission this data was merely illustrative; we have now expanded figure 2 to include a quantitative analysis of spike entrainment to the optogenetic stimulation and also amended the text, p.7, ll.165-168, and the methods, p.25, ll.624-628. The information related to which animals exhibited entrainment has also been included in the newly added table 3, p.24, l. 616

(4) The low optogenetically evoked MUA responses especially in V1 are surprising. Could the authors show the alignment of the RFs mapped for the optogenetic injection site and for the subsequent cortical recording sites for the animals and link them to the results in each animal, please?

We attempted to find optimal RF alignment between LGN and V1 RFs using manual mapping and did not perform quantitative mapping during this phase of the experiment. We are therefore unable to present the data requested; we attribute the relatively weak MUA responses in V1 to the -relatively – lower probability of finding synaptically coupled V1 units compared to the higher probability of seeing mass action signals in the field potentials. We have added a sentence related to this point in the Results section when describing Figure 6, p.13, ll. 288-289.

(5) Figure 2B figure legend: add that this is visual stimulation, please.

We have added this information.

(6) From which animal is the performance shown in Figures 3A and 4B? Please add to the figure or figure legend.

We have added this information.

(7) sFigure 6: For animal 1409, it would be helpful to explain to the reader how a performance of that is generally below 50% correct leads to a d' >1 on the first day of optogenetic stimulation. Unless I am misunderstanding the two graphs?

We thank the reviewer for identifying this issue, which was due to an analysis error on our part. The correct data is now shown in supplementary Figure 6, and reveals elevated %correct results that underlie the d’ value > 1 on the first day. This error affected two of the three animals; and indeed now the learning curves reveal transfer task acquisition during the first tens of trials in these animals.

Reviewer #3 (Recommendations for the authors):The paper is much improved, and the focus on the behavioral effects of LGN optogenetic stimulation strengthens the paper. While the basic finding seems to be robust, some details are not described adequately, several aspects of the presentation can be improved, and the manuscript could benefit from more careful editing.Specific comments– Figure 2 is nice but this is only one example. It is important to include a summary figure to follow up on Figure 2 that shows quantitatively how key findings from Figure 2 generalize to all recorded cells. For example, do most cells phase lock up to 60 Hz? What is the overall firing rate as function of stimulation frequency?

In the original submission this data was merely illustrative; we have now expanded figure 2 to include a quantitative analysis of spike entrainment to the optogenetic stimulation, and also amended the text, p.7, ll.165-168, and the methods, p.25, ll.624-628. The information related to which animals exhibited entrainment has been included in the newly added table 3, p.24, l. 616.

– Please change y–axis of PSTHs in Figures2 and 6 to firing rate.

Done.

– Figure 4B – should be mW and not W in the key.

Done.

– In Figure 4 it would be helpful to add an icon or text in B to indicate which animals are controls.

Done.

– In Figure 5A, it would be helpful to add a panel that shows d' per session as a function of session # for each animal. The session–to–session variability is important to assess in order to interpret the effects of power and frequency in Figure S6 since each combination was only used once per animal.

We feel that the variability is already captured by the SEM in this figure and additional panels would clutter the figure.

– Figure 6 is missing panel labels.

We have corrected this omission.

– 5 units in V1 showing significant change in firing rate during optostim at p<0.05 out of 111 is expected by chance and the modulation according to Figure 6A is very modest; a more appropriate description of these results is that the authors failed to see V1 effects in the single/multi–unit level.

Technically the reviewer is correct; however looking at the histograms and raster plots these responses, although infrequent in LGN and rare in V1, do not appear to result from random statistical fluctuations. It is not a main point of the paper, but we prefer to keep this finding in the manuscript.

– In Figure 8B, each set of connected symbols is a different site. If so, can you comment on why the effect is so bimodal? Are filled symbols significant optostimulation effect?

We already discuss this in the Results section; this apparent bimodality is most likely related to RF overlap.

– Why compute STA of LFP in V1 based on LGN spikes rather than on based LGN optogenetic stimulation? Is there a significant difference between these two?

We compute the STA using LGN spikes as the optogenetic induced LGN spiking activity will have a more direct effect on cortical processing that the optogenetic LGN stimulation.

– Was there any indication to the animal of a miss other than withdrawal of the reward? Was the intertrial interval fixed? Was the ITI affected by misses?

There was no explicit cue delivered upon a missed stimulus.

– The text discussing "on" and "off" cells is a little confusing. It would be better to say explicitly that "on" and "off" visually responsive cells are segregated in the tree shrew LGN layers and refer to the supplementary figure.

We have added “visually responsive” to the paragraph in the discussion as suggested by this reviewer, p.17, l. 398.

– The terminology may be confusing to the readers. Instead of "light stimulation", perhaps say "visual stimulation" to better contrast with optogenetic stimulation?

We indeed use the term visual stimulation to refer to sensory input through the retina, whereas “light stimulation” and “optogenetic stimulation” refer to activation of neurons by light delivery to the brain.

– It would be helpful to indicate in the figures which panels show results from optogenetic stimulation and which from visual stimulation.

We have now indicated this in the figure legends.

– "as has been reported also for humans (Erickson–Davis and Korzybska, 2021)" may be interpreted by the reader as implying that optogenetic stimulation has been done in humans but this is an electrical stimulation study.

We have added “in work using electrical stimulation”, p.11, l.249 as suggested by the reviewer.

– "it has been shown that for macaques, extensive training on optogenetic stimulus detection in V1 can be detrimental for detection of visual stimuli (Ni and Maunsell, 2010)" – this paper used microstimulation and not optogenetic stimulation.

The reviewer is right and we have changed this to “detection of electrical microstimulation”, p.16, l.376